# Synthesizing Feature Extractors: An Agentic Approach for Algorithm Selection

## Abstract

Algorithm selection for constraint satisfaction problems requires extracting features that capture problem structure. Manually designing feature extractors demands deep domain expertise and becomes a bottleneck when facing new problem classes. We present an automated approach using Large Language Models to synthesize executable Python scripts that function as interpretable feature extractors. Given a high-level constraint model in MiniZinc, an LLM agent generates code that constructs a typed graph representation and computes structural properties, such as graph density, variable clustering, and constraint tightness. We validate our approach on algorithm selection across 227 combinatorial problem classes from MiniZinc Challenges (2008–2025). Our synthesized extractors achieve 58.8% accuracy versus 48.6% for human-engineered extractors (mzn2feat), and outperform neural baselines by 6.8 percentage points on FLECC and 4.3 points on Car Sequencing while maintaining full interpretability. This demonstrates that program synthesis can automate feature extraction for constraint optimization without sacrificing transparency.

## 1 Introduction

Consider the vehicle routing problem (VRP): a delivery company must route trucks to customers, respecting capacity and time constraints while minimizing travel distance. These problems are computationally hard; solvers can run for hours without finding proven optima, yet even small improvements in route efficiency translate to substantial cost savings at scale. The difficulty arises from the structure of the problem. Two VRP instances may differ in customer density (clustered vs. scattered), time window tightness, or demand distribution. These structural characteristics determine which solver performs best: mixed-integer programming excels on instances with certain structures, while constraint programming or hybrid approaches work better on others.

Understanding and exploiting such structural characteristics is fundamental to solving combinatorial optimization problems efficiently. By identifying these characteristics as features, such as graph density, constraint tightness, and variable dependencies, we can predict computational hardness and select the most suitable algorithmic approaches. Since the 1970s, algorithm selection has emerged as a cornerstone technique in machine learning (ML) and optimization, promising to match each problem instance with its most effective solver from a portfolio of complementary algorithms (Rice, 1976).

Yet a critical bottleneck remains: designing effective feature extractors. Creating a tool that compiles a problem instance into a succinct and representative set of features requires deep domain expertise, intimate knowledge of which features are most important for algorithmic performance, and efficient extraction methods that do not consume excessive computational resources. This challenge becomes particularly acute when facing new problem domains. Researchers from different problem domains find it difficult to leverage algorithm selection techniques, simply because no feature extractor exists for their specific problem Xia & Szeider (2024); Kerschke et al. (2019).

The conventional workaround—translating the problem into another formalism for which extractors exist—often proves inadequate. When we compile a high-level problem description into a flat constraint representation, we lose crucial structural information that was explicit in the original formulation. Global properties and graphical relationships that are immediately apparent in the high-level description become obscured or entirely invisible once the problem is flattened. This loss of information directly impacts the quality of algorithm selection, as the features extracted from the flat representation fail to capture characteristics that differentiate easy instances from hard ones.

**Our Approach: Automating Feature Extraction via LLMs** This paper presents a novel approach that fundamentally changes how feature extractors are created. Rather than requiring manual engineering by domain experts, we introduce an LLM-based framework that automatically generates executable Python scripts serving as feature extractors. Our key insight is a *two-level process*. We first use a Large Language Model via an agentic error correcting workflow to generate an executable program. This program then serves as the feature extractor, capturing the essential characteristics of the problem.

Our approach operates on high-level problem descriptions written in MiniZinc (Stuckey et al., 2014; Marriott et al., 2008), a declarative modeling language for constraint satisfaction and optimization problems. This choice is deliberate: rich, expressive formalisms enable more compact problem representations, making it easier for the LLM to identify and exploit structural patterns. The LLM agent analyzes the MiniZinc model and generates a complete Python script that, when executed on a problem instance, produces a graph representation of the problem instance, which in turn yields a vector of interpretable features.

Critically, our framework produces explicit, interpretable features rather than opaque neural embeddings. While recent work has explored using deep learning to create latent representations of problems (Pellegrino et al., 2025; Zhang et al., 2024; Loreggia et al., 2016), such approaches often sacrifice transparency for the sake of automation. In contrast, our generated extractors produce features that domain experts can understand, validate, and refine, including graph density, variable clustering coefficients, constraint tightness, and statistical properties of the data. This "gray box" approach ensures the automated extraction process remains accessible to human understanding and improvement.

**Two Complementary Frameworks** We develop two distinct but complementary frameworks for feature extraction.

The *problem-specific framework* generates tailored extractors for individual problem types, analyzing the particular structure and semantics of problems like *vehicle routing problem* (VRP) or *car sequencing* (CS). In this framework, the LLM agent is provided with the high-level problem description, instance data, and schema information. From these inputs, it generates a specialized Python script that extracts, via the construction of a custom graphical representation of the instance, approximately 50 characteristics relevant to algorithm selection for that particular problem.

The problem-generic framework takes a more ambitious approach: applying a universal feature extractor to *any* constraint satisfaction problem. The key innovation here is the use of a *universal* intermediate graph representation (in contrast to a custom one in the problem-specific framework). The LLM-generated script first converts any problem instance into a standardized bipartite graph, where nodes represent variables, constraints, and weighted edges encode their relationships. From this graph representation, we extract a uniform set of structural features using standard graph analysis algorithms. This two-level approach, where the LLM generates a script that converts the problem into a graph, from which features are then extracted, preserves high-level structural information while enabling consistent feature extraction across diverse problem types.

**Empirical Validation and Surprising Results** To validate our approach, we conducted experiments in algorithm selection scenarios using a portfolio of five leading solvers: Gurobi, CPLEX, SCIP, Gecode, and OR-Tools. For the problem-specific framework, we evaluated performance on three distinct problem types with sufficient instance diversity. For the problem-generic framework, we utilized a benchmark comprising over 2,000 instances spanning 227 different problems from two decades of MiniZinc Challenges (2008-2025).

Our automatically-generated extractors consistently and substantially outperform *mzn2feat*, the established feature extractor for MiniZinc problems (Amadini et al., 2013; 2014). Algorithm selectors trained on our features achieved 58.8% accuracy on the generic benchmark suite, compared to 48.6% for those using *mzn2feat* features, a notable improvement that held across different ML models and problem types. This performance gain stems from our extractors' ability to capture high-level structural properties that remain hidden in flat representations.

**Contributions and Significance** This work makes several significant contributions to combinatorial optimization and algorithm selection:

1. We demonstrate that LLMs can effectively reason about combinatorial problem structure and generate functional feature extractors wth minimal human guidance. This automation

democratizes access to algorithm selection techniques, enabling their application to new problem domains where manual feature engineering would be prohibitively expensive.

2. We introduce a novel approach that utilizes LLM-generated scripts as an intermediate representation, preserving interpretability while achieving automation. Unlike end-to-end neural approaches, our framework produces extractors that humans can understand, validate, and improve upon, facilitating human-AI collaboration in algorithm design.

3. Through comprehensive empirical evaluation, we show that automatically generated extractors can surpass carefully engineered alternatives, suggesting that LLMs can identify subtle structural patterns that human experts might overlook. This finding has immediate practical implications for building more effective algorithm portfolios and enhancing the efficiency of combinatorial problem solving.

4. Our framework operates within reasonable computational budgets (minutes cost for extractor generation), making it accessible to researchers and practitioners without extensive resources. The generated extractors themselves are lightweight, adding minimal overhead to the algorithm selection pipeline.

**Paper Organization** The paper is organized as follows: we first introduce preliminaries and related work, and then present the two frameworks. Next, we present the experimental results and analysis, and finally conclude the paper by outlining promising further directions.

## 2 RELATED WORK

The *Algorithm Selection Problem* (AS) (Kerschke et al., 2019) considers a portfolio $\mathcal{P}$ of algorithms, a set of problem instances $I$, a performance metric $PM(A, i)$, and a resource budget $B$. Since the performance of an algorithm $A \in \mathcal{P}$ typically varies across instances, an AS strategy must predict, *before solving*, which $A$ to run on a given instance.

To make such predictions tractable, each instance $i \in I$ is described by a *feature vector* $\phi(i) \in \mathbb{R}^d$ obtained by a feature extractor $\Phi : i \mapsto \phi(i)$. The AS task is then to learn a selector $S : \mathbb{R}^d \to \mathcal{P}$ that, given $\phi(i)$, chooses an algorithm $S(\phi(i)) \in \mathcal{P}$. The objective is:

$$\max_{S \text{ respecting } B} \sum_{i \in I} PM\big(S(\phi(i)), i\big).$$

MiniZinc (Nethercote et al., 2007) is a high-level, declarative modeling language used to describe constraint satisfaction and optimization problems in a solver-independent way. A MiniZinc model file (.mzn) defines the problem, including variables, constraints, and optionally an objective (e.g., to minimize or maximize a specific value). A MiniZinc data file (.dzn) provides input data for the model, specifically assigning values to parameters declared in the model file. Other related definitions are in Appendix A 11.

Feature engineering for machine learning involves creating or selecting features from data to enhance model performance through feature selection (identifying relevant features), feature extraction (applying transformations to reduce dimensionality), and feature synthesis (generating new features from existing ones).

**LLM-Based Feature Engineering.** CAAFE (Hollmann et al., 2023) uses LLMs to generate Python scripts that transform tabular ML datasets into augmented feature sets, with each generated feature accompanied by explanatory comments. CAAFE operates on traditional ML datasets with a tabular structure. Our work addresses a different input type: constraint optimization problems defined by model files (.mzn) and data files (.dzn). We extract structural and semantic properties from declarative problem specifications rather than transforming existing feature vectors.

**Neural Embeddings for Algorithm Selection.** Wu et al. (Wu et al., 2024) use LLMs to generate algorithm embeddings by processing algorithm source code and documentation, producing dense representations for algorithm selection. Zhang et al. (Zhang et al., 2024) combine graph neural networks with expert knowledge for SAT solver selection, learning embeddings from CNF formula structure. Pellegrino et al. (Pellegrino et al., 2025) apply transformer architectures (*trans2feat*) to learn features from constraint optimization problems. These neural approaches produce high-dimensional embeddings where individual dimensions lack clear semantic meaning and cannot be easily inspected or modified by domain experts. On the FLECC and Car Sequencing benchmarks used by Pellegrino et al., our LLM2feat-based selectors outperform all *trans2feat* variants by 6.8 and 4.3 percentage points in test accuracy, respectively, while requiring no per-problem retraining (detailed comparison in Appendix C, Tables 13-16).

**Our Contribution: Interpretable Program Synthesis.** Our framework generates explicit, interpretable features as executable Python scripts. Domain experts can inspect the chosen features, validate their correctness, and refine the generated extractors. This differs from end-to-end learned representations, where such inspection and modification are not possible. Our problem-generic framework synthesizes feature extractors applicable to any constraint optimization problem that can be expressed in MiniZinc. We evaluate on 227 problem classes, compared to prior work targeting specific problem types (SAT formulas in (Zhang et al., 2024), three constraint problems in (Pellegrino et al., 2025), tabular data in (Hollmann et al., 2023)).

## 3    ALGORITHM FRAMEWORKS

In this section, we present the frameworks for our two LLM-based agents, which are inspired by the recent agentic framework that bridges LLM and constraint satisfaction problem (CSP) solving (Szeider, 2025). Our LLM-agent system takes a problem instance (including a .mzn and a .dzn file) in the MiniZinc language as input and outputs a Python script that extracts relevant features from the input problem instances.

### 3.1    PROBLEM-SPECIFIC LLM-BASED AGENT

For the specific-problem framework, the agent problem-related files (minizinc model file .mzn and problem instance file .dzn) and a data schema are required, which can provide the LLM with a detailed understanding of the input data structure. Meanwhile, we have two prompts: (script_system_prompt and mzn-tuning) controlling the generation of feature extractors (a Python program) in an efficient way.

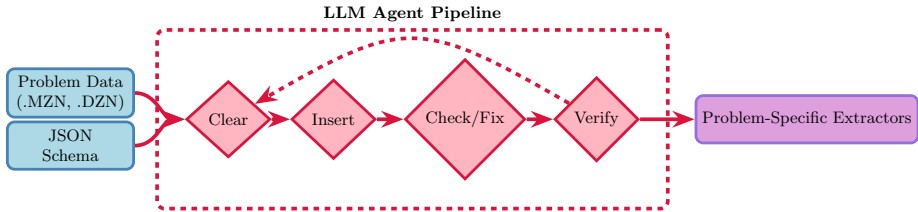

Figure 1: The workflow of generating problem-specific feature extractors

script_system_prompt is a general-purpose Python script providing behavioral instructions for LLM agents to generate complete, executable Python scripts through a tool-based environment. It defines a strict workflow, technical requirements, and available resources. The example of the general Python script template structure is in the appendix. The general-purpose script generation prompt follows a structured tool-based workflow, and its procedure follows Figure 1:

1. **Clear**: Clearing all previous content in the Python script.
2. **Insert**: Creating complete Python script.
3. **Check/Fix**: Verifying the syntax and requirements from the prompts; Addressing validation errors if needed.
4. **Execute**: Running the Python script and capturing and validating the output

The mzn-tuning prompt contains specialized instructions for LLM agents to generate constraint optimization feature extraction scripts. Unlike the general script system prompt, this is highly domain-specific for AS, guiding the agent to generate Python scripts that extract standardized instance features from constraint programming problems to train algorithm selectors for optimal solvers. The feature extractor template specifically for the Minizinc instance is in the appendix. The specialized prompt follows a research-oriented feature extraction workflow:

1. **Template Substitution**: Replace placeholders `${INSTANCE}` (.dzn file), `${SCHEMA}` (the prompt helping LLM understand the .mzn model file), `${MODEL}` (.mzn file), `${PROBLEM}` (problem name).
2. **Mandatory Imports**: Exact import requirements for framework integration
3. **Data Access**: Use `input_data()` - no file I/O operations
4. **Feature Analysis**: Extract 50 standardized features
5. **Testing**: `execute_script()` validation required
6. **Output**: Standardized format for algorithm selector training

### 3.2    PROBLEM-GENERIC LLM-BASED AGENT

In a further step, we design a generic workflow that works on multiple cross-domain problems simultaneously. The framework is a generic problem-generic pipeline where the output Python

scripts include not only a parser and a feature extractor, like the problem-specific pipeline, but also a generic graph builder. After the parser handles and processes the input data, the generic graph builder constructs a graph with different types of nodes (like variables, constraints, and so on), and weighted edges representing their relationships. Based on the generic graph, the feature extractor applies graph analysis to uniformly extract structural features, obtaining the same features for diverse CP problems. Figure 2 illustrates the monolithic architecture of the problem-generic pipeline for generating feature extractors. It has a similar overall framework to the problem-specific one, but we use a subagent to generate Python scripts (converters) that integrate graph builders. Both frameworks use the same check-fix-verify loop shown in Figure 1; Figure 2 includes the feedback from verification back to the LLM inside the subagent pipeline.

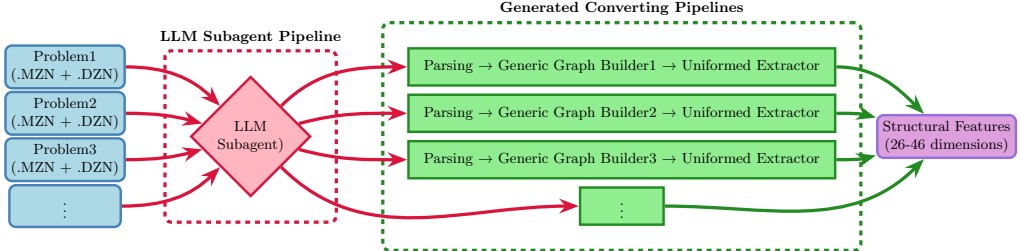

Figure 2: The workflow of generating problem-generic feature extractors

## 4 EXPERIMENTAL ANALYSES

### 4.1 EXPERIMENTAL SETTINGS

In an AS problem, the goal is to select the most suitable (best) solver from a pool of candidates to solve a given MiniZinc instance, which consists of a model file and a corresponding data file. The pool $\mathcal{P}$ of solvers is made up of the best performing solvers in the Minizinc challenge [1]: Gurobi (12.0.3), CPLEX (22.1.2), SCIP (9.2.3), Gecode (6.2.0), and OR-Tools (9.3.10497). This portfolio was selected to provide broad coverage across solver paradigms: Gurobi and CPLEX are commercial mixed-integer programming (MIP) solvers used in industrial applications and MIPLIB benchmarks; SCIP combines CP and MIP techniques; Gecode is a constraint programming (CP) solver used throughout the MiniZinc Challenge history; and OR-Tools won gold medals in the MiniZinc Challenge across all major categories in 2023, 2024, and 2025. The set of instances $I$ includes minimization/maximization problems, where the best solver means achieving the lowest/highest objective values after the timeout, and decision problems, where the best solver means the shortest solving time to get the results. In particular, for the experiments of the problem-specific framework, we extract features and train algorithm selectors on three problems: VRP (Queiroga et al., 2021), CS (Pellegrino et al., 2025), and fixed-length error-correcting codes (FLECC) (Pellegrino et al., 2025), where we obtain sufficient instances for training and testing. In the experiments of the problem-generic framework, we comprehensively collect all MiniZinc Challenges (Stuckey et al., 2014) [2] from 2008 to 2025, where we have more than 2000 instances spanning 227 problems of combinatorial optimization after filtering out the problems with fewer than three instances. For both scenarios, problem-specific and problem-generic frameworks, we split the instances randomly into training and test sets by 7 : 3 ratio. To ensure fairness in each pair of comparisons, we use the same split of training and test sets.

We run solvers with a 20 mintues timeout to get the evaluation. We run the LLM pipeline to generate with *LLM2feat* extractors with a 1 minute timeout. We run all the instances on all solvers on a compute cluster with nodes equipped with two AMD 7403 processors (24 cores at 2.8 GHz) and 32 GB of RAM per core. The performance metric is $PM(A, I)$ where $A$ is the solver and $I$ is the set of instances, and $PM(A, I)$ is the number of instances for which solver $A$ is the best solver. Consequently, we have two metrics: the ratio of selecting the best solver ($Acc$), and the average ranking of selected solvers ($Rank$) for the problem-specific evaluation. An additional Borda score from the competition [3] is used for cross-problem evaluation in the problem-generic framework. It generally means how many other solvers a solver can outperform.

---

[1]https://www.minizinc.org/challenge/2025/results/

[2]https://www.minizinc.org/challenge/

[3]http://www.minizinc.org/challenge/2025/

For the selection of LLM models, we use OpenAI o4-mini-2025-04-16 and Anthropic claude-sonnet-4-20250514 in the problem-specific and problem-generic framework, respectively. Details regarding the LLM model's sensitivity and selection can be found in the Appendix. Regarding the necessity of the agentic workflow components, we found that all parts of the check-fix-verify loop are essential: attempts to skip checking, fixing, or verification steps consistently resulted in failure to generate executable feature extractors.

## 4.2 RELATED TOOLS FOR ALGORITHM SELECTION

The input to the AS tools includes a feature table containing problem instances with their corresponding features generated by feature extractors, as well as a performance table that includes the instances and the performance metrics of different solvers when solving these instances. To get the features of the instances we use our LLM-based frameworks (*LLM2feat*) and *mzn2feat* as feature extractors.

In particular, we use the following AS tools in our experimental analysis:

Multiclass classification models for predicting the best solver. We use Random Forest (referred to as RF) in the standard way (Kerschke et al., 2019), and AutoSklearn (Feurer et al., 2015; 2020) (referred to as AutoSK). Their corresponding parameter settings are listed in the Appendix. We use two loss functions: accuracy, $Acc$, and average ranking, $Rank$. By accuracy, we mean the ratio of times the model predicts the actual best solver, and by average ranking, we mean the average rank of the solver the model predicts. For all training, we use the cross-validation with 5 folds.

AutoFolio (Lindauer et al., 2015) (referred to as AF) and LLAMMA (Kotthoff, 2013) (referred to as LLAMMA), which are the best performing AS tools as reported in surveys (Kerschke et al., 2019) and competition [4]. For these tools, only the accuracy loss function is used for training, as we utilize the tools in their default settings.

The combination of the above AS tools and feature extractors yields the following list of AS approaches: *mzn2feat*+AF; *mzn2feat*+RF; *mzn2feat*+LLAMMA; *mzn2feat*+AutoSK; *LLM2feat*+AF; *LLM2feat*+RF; *LLM2feat*+LLAMMA; *LLM2feat*+AutoSK.

## 4.3 GOALS OF THE EXPERIMENTAL ANALYSES

We want to answer the following questions by extensive experiments:

- Q1: How diverse are features generated by *LLM2feat*? Ideal features in the same set are expected to be informatively diverse. Highly correlated features provide overlapping information, reducing the effective dimensionality of the feature space. Diverse features enable ML models to possibly capture complementary problem features and different aspects of problem complexity with fewer parameters.
- Q2: How efficiently does each feature contribute to the algorithm selection models? By analyzing the importance of features, we can generally understand the quality of the features and identify potential opportunities for reducing redundant feature sets.
- Q3: How accurate are the AS models using *LLM2feat* and *mzn2feat* features? Applying the feature sets to AS models can directly validate the usefulness of these features for AS.

## 4.4 PROBLEM-SPECIFIC FEATURE EXTRACTOR

For our problem-specific feature extractors, we analyze feature sets extracted by *mzn2feat* and *LLM2feat* extractors across three different problems through an extensive series of experiments. Firstly, we check the correlation between the features in both sets and the general diversity of the feature sets; then we check the features' importance and quality of the features generated by *mzn2feat* and our *LLM2feat*; Finally, we verify the effectiveness by training algorithm selectors with *mzn2feat*-based and *LLM2feat*-based features and compare their corresponding accuracy.

### 4.4.1 FEATURE CORRELATION ANALYSIS

Given a feature set, we can train a standard algorithm selector using random forests as described in 4.1. Consequently, we obtain two algorithm selectors, *mzn2feat* + RF and *LLM2feat* + RF, from which we can get the top 20 features of the node that contribute the most to the classification decision. This approach ensures that our correlation analysis focuses on algorithm-selection-relevant features rather than examining all features indiscriminately. For each feature set, we computed Pearson correlation matrices (Guyon & Elisseeff, 2003) and analyzed both the distribution of correlation

---

[4]https://www.coseal.net/open-algorithm-selection-challenge-2017-oasc/

coefficients ($r$) and their statistical properties. We present the heatmap figures of correlation coefficients from two feature sets for VRP problems (the results for FLECC and CS problems are in the subsection of the Appendix). LLM features show the most substantial diversity advantage with an average correlation magnitude of $|r| = 0.221$ compared to *mzn2feat*'s $|r| = 0.429$. For the other two problems, the tendency is also the same: these results demonstrate that LLM-generated features capture more diverse aspects of problem structure, reducing information redundancy and enabling more efficient representation of constraint optimization characteristics with fewer overlapping features.

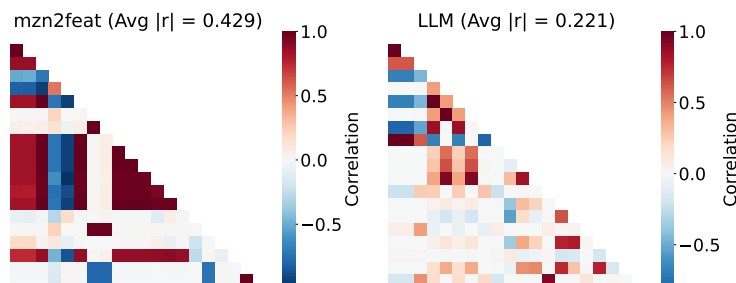

Figure 3: Feature correlation analysis for VRP problem (feature names removed for clarity). LLM features exhibit the largest diversity advantage with $48.5\%$ lower average correlation ($|r| = 0.221$) compared to *mzn2feat* ($|r| = 0.429$).

### 4.4.2 FEATURE UTILIZATION EFFICIENCY

Beyond correlation analysis, we examine the importance of each feature within feature sets. This analysis addresses Q2 by evaluating the distribution of feature importance across all available features in each dataset.

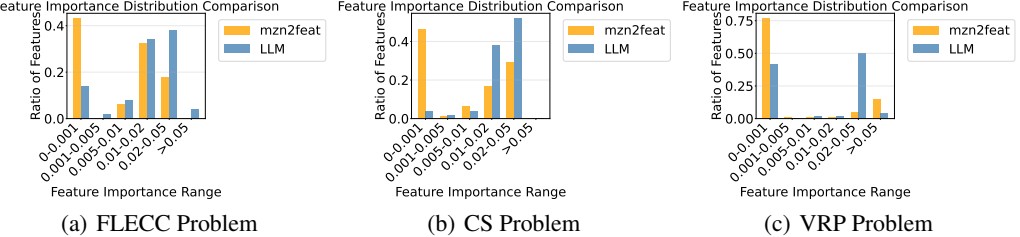

(a) FLECC Problem        (b) CS Problem        (c) VRP Problem

Figure 4: Feature importance distribution analysis across three constraint optimization problems. Consistent patterns show LLM features achieve better distribution across importance ranges.

We analyzed feature utilization efficiency using a comprehensive approach that examines the complete feature set rather than focusing solely on top-performing features. For both *mzn2feat*+RF and *LLM2feat*+RF, we extracted feature importance scores for all features and applied a significance threshold of $0.001$ to identify "effectively utilized" features. This threshold ensures we capture features that contribute meaningfully to algorithm selection decisions while filtering out noise. Our analysis encompassed all 95 human-crafted features of *mzn2feat* and 50 *LLM2feat*-based features. The problem-specific framework produced features that demonstrate consistently superior utilization efficiency across all three problems. From Figure 4, for example, on the VRP problem, LLM achieves $96\%$ utilization efficiency compared to *mzn2feat*'s $56.8\%$, representing a $69\%$ improvement in effective feature usage. On FLECC, LLM reaches $58\%$ efficiency versus *mzn2feat*'s $23.2\%$; On CS problem, LLM attains $84\%$ efficiency compared to *mzn2feat*'s $45.1\%$.

### 4.4.3 ACCURACY ANALYSIS

To directly evaluate the practical impact of our feature extraction approaches, we conducted a comprehensive accuracy analysis that addresses Q3. This analysis examines the performance of algorithm selection as a function of feature set size, providing insights into both the effectiveness and efficiency of LLM-generated features versus traditional features.

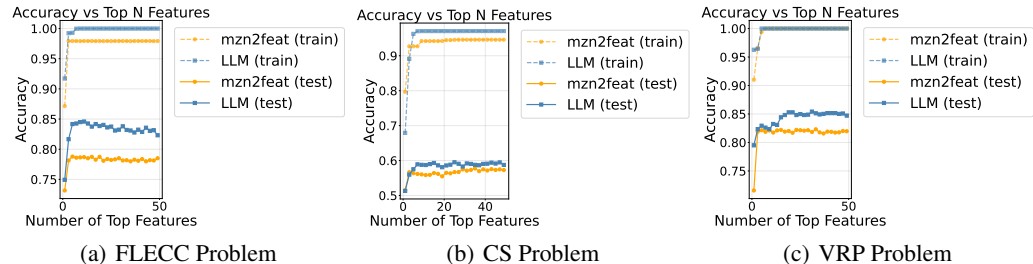

(a) FLECC Problem      (b) CS Problem      (c) VRP Problem

Figure 5: Accuracy analysis across three constraint optimization problems demonstrating consistent LLM superiority in algorithm selection performance and feature efficiency.

We implemented a systematic feature scaling analysis that evaluates algorithm selection accuracy using incrementally expanding feature sets. Starting with the single most important feature from each extraction method, we progressively added features in the order of decreasing importance, testing every other feature count from 1 to 50 features for computational efficiency. For each feature subset size, we selected the top-N features based on Random Forest feature importance rankings and re-trained the algorithm selectors using only these selected features. Finally, we evaluated performance on both training and testing sets. From Figure 6, we can see LLM-based features reach near-optimal performance with greater accuracy with different numbers of features. Meanwhile, *LLM2feat*-based algorithm selectors have higher chances to improve the accuracy when more features are given, like on the VRP problem, when there are more than 10 features, *mzn2feat*-based features do not help for further improvement (the accuracy is always around $81\%$), but *LLM2feat*-based features can reach the highest values when more than 20 features are given.

To verify that *LLM2feat* provides more informative structural features than *mzn2feat*, we also use respective features for training the algorithm selector with other ML tools (AF, LLAMMA, and AutoSK) besides RF for the three problems. From Table 1, we can see that, except when features are fed to AF, which always yields the same results as the single best (SB, the definition refers to the Appendix A.11), the algorithm selectors based on *LLM2feat* consistently achieve higher accuracy and better ranking (lower ranking indicates better performance) on the test set. Therefore, *LLM2feat* can always get more informative problem feature sets for ML tools to utilize to select better solvers. In the Appendix, we also present the results on the training set using $Acc$ as the loss function, as well as the results on both the training and test sets with $Rank$ as the loss function.

| Loss function=$Acc$ (Testing) | VRP | | CS | | FLECC | | Suite (227 problems) | | |
|---|---|---|---|---|---|---|---|---|---|
| | Acc | Rank | Acc | Rank | Acc | Rank | Acc | Rank | Borda |
| SB | 79.5% | 1.221 | 49.0% | **1.616** | 78.2% | 1.420 | 27.0% | 2.667 | 0.694 |
| *mzn2feat*+AF | 79.5% | 1.221 | 49.0% | **1.616** | 78.2% | 1.420 | 27.0% | 2.667 | 0.694 |
| *mzn2feat*+RF | 83.0% | 1.184 | 58.5% | 1.680 | 79.5% | 1.391 | 47.4% | 1.419 | 1.323 |
| *mzn2feat*+LLAMMA | 82.9% | 1.184 | 59.4% | 1.798 | 78.7% | 1.397 | × | × | × |
| *mzn2feat*+AutoSK | 84.4% | 1.166 | 62.1% | 1.787 | 79.4% | 1.394 | 48.6% | 1.464 | 1.320 |
| *LLM2feat*+AF | 79.5% | 1.221 | 49.0% | **1.616** | 78.2% | 1.420 | 27.0% | 2.667 | 0.694 |
| *LLM2feat*+RF | 85.7% | **1.155** | 61.3% | 1.681 | 83.5% | 1.338 | 58.3% | **1.340** | **1.426** |
| *LLM2feat*+LLAMMA | 85.8% | 1.157 | 61.0% | 1.773 | 84.2% | **1.306** | × | × | × |
| *LLM2feat*+AutoSK | **85.4%** | 1.160 | **64.0%** | 1.738 | **84.6%** | 1.310 | **58.8%** | 1.390 | 1.421 |

Table 1: The accuracy and the average ranking results of *mzn2feat* features (95) and *LLM2feat* features (50) trained with different models on the testing set (loss function= $Acc$). Note: the results of the problem-generic framework on the problem suite, including 227 problems, are also listed here, and they will be discussed later. LLAMMA is not applicable to cross-problem algorithm selection, so it is masked.

### 4.4.4 QUALITATIVE FEATURE EXAMPLES

To illustrate the solver-aware nature of LLM-generated features, consider the VRP problem: the LLM automatically synthesized features such as demand distribution statistics (`avg_demand`, `std_demand`, `skew_demand`, `coeff_var_demand`), depot centrality metrics (`depot_centrality`, `depot_bc`), and MST-based route bounds (`mst_total`, `mst_avg_edge`). These features enable the selector to distinguish between cases where MIP solvers (Gurobi, CPLEX, SCIP) excel versus those better suited to CP solvers (OR-Tools,

Gecode) (Moreno-Scott et al., 2016), explaining the $+2.7$ percentage-point accuracy gain over *mzn2feat* on VRP. Additional qualitative analysis and feature categorization are explained in Appendix B, Qualitative Feature Analysis.

### 4.5 PROBLEM-GENERIC FEATURE EXTRACTOR

For the more general framework, the problem-generic framework, we test them in a more systematic way on 227 problems. We also analyze the feature correlation, feature utilization efficiency, and accuracy analysis as we do for the problem-specific framework. The outperformance of the features generated by *LLM2feat* is even more significant compared with that generated by *mzn2feat*.

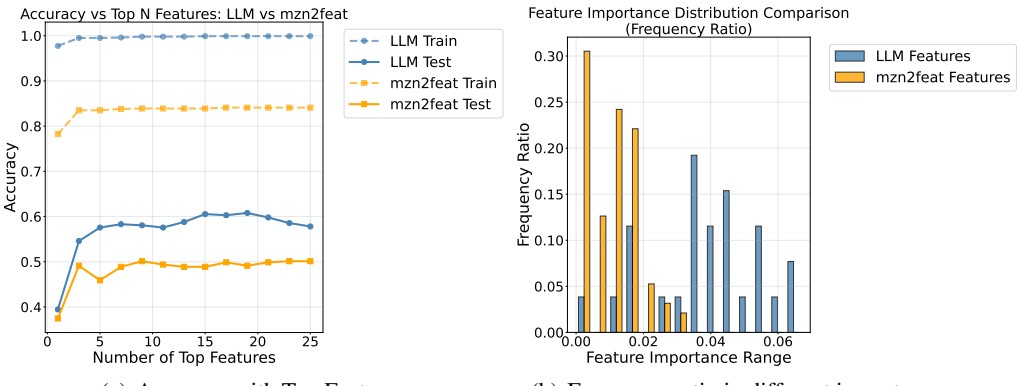

(a) Accuracy with Top Features      (b) Frequency ratio in different importance range

Figure 6: Accuracy analysis across 227 constraint optimization problems demonstrating consistent LLM superiority in algorithm selection performance, and informative features obtained by LLM than *mzn2feat*

For feature correlation, LLM features show significantly lower average correlation ($|r| = 0.141$) compared to *mzn2feat* ($|r| = 0.245$), demonstrating the improvement in feature diversity (The heapmap figure is in the Appendix). From the feature importance distribution comparison in Figure 6, the feature utilization of LLM-based features is more efficient as *LLM2feat* generates more features with higher importance than *mzn2feat*. On the test accuracy with the top important features, the *LLM2feat*-based RF always gets higher accuracy than the *mzn2feat*-based RF, which means given the same number of important features from respective feature sets, *LLM2feat*-based features are more informative for training algorithm selectors. For example, with around 15 top features, *LLm2feat*-based RF can get $60\%$ on the test set, which is much higher than the accuracy ($49\%$) obtained by *mzn2feat*-based RF.

From Table 1, we have the results of different metrics obtained by various *LLM2feat* and *mzn2feat*-based algorithm selectors: LLM-based algorithm selectors can achieve at least $10\%$ higher accuracy, and better ranking. Moreover, we add another evaluative metric, Borda scores, which also indicates *LLM2feat*-based algorithm selectors can also get higher scores than their competitors.

## 5 CONCLUSION AND FUTURE WORK

To the best of our knowledge, we present the first LLM-based framework that synthesizes graph-theoretic, interpretable feature extractors from symbolic constraint models (MiniZinc) for algorithm selection across hundreds of combinatorial problems. The framework is grounded in the use of LLM agents, which are parametrically instructed to accommodate potentially diverse representation formalisms. The extractors are generated as Python scripts that can subsequently be refined by human experts in a "gray-box" manner. The construction process is computationally efficient and can therefore be employed on demand for dynamic scenarios. Moreover, we have demonstrated the effectiveness of these extractors in a representative ML task—algorithm selection—showing that they can achieve competitive performance even in domains where human-engineered feature extractors are already available.

REPRODUCIBILITY STATEMENT

We took several steps to make our results easy to reproduce.

**Code and artifacts.** We release an anonymized supplement for review containing: (i) the full training/evaluation pipeline for all selectors, (ii) the LLM prompting templates and the exact prompts used, (iii) all LLM-generated feature-extractor scripts (with content hashes).

**Data and benchmarks.** All experiments use public MiniZinc Challenge benchmarks (2008–2025). We provide scripts to download the models and instances, and a manifest listing the subset used after filtering problems with fewer than 3 instances. We include our fixed train/test split (70/30) as instance lists to ensure identical data partitions across runs.

**Solvers and versions.** We report and pin solver versions in all runs: Gurobi 12.0.3, CPLEX 22.1.2, SCIP 9.2.3, Gecode 6.2.0, OR-Tools 9.3.10497. We provide wrapper scripts that uniformly handle time limits, seeds, and logging. Proprietary solvers (Gurobi/CPLEX) are optional; our scripts fall back to open-source solvers (SCIP, Gecode, OR-Tools).

**LLM configurations.** For problem-specific extractors we used OpenAI `o4-mini-2025-04-16`; for the problem-generic pipeline we used Anthropic `claude-sonnet-4-20250514`. We release: (a) the system and task prompts, (b) the template files, (c) the top-level orchestration script, and (d) the raw generated extractors. We cap extractor generation to 60 s per script and keep all retries; each artifact is identified by a SHA-256 hash.

**Learning pipelines and hyperparameters.** We evaluate Random Forest (RF), AutoSklearn (AutoSK), AutoFolio (AF), and LLAMA with default settings unless specified.

**Evaluation protocol.** For each instance we run all portfolio solvers with a 20-minute wall-clock limit and parse outcomes into a unified performance table. Selector models are trained on the training split only; metrics are reported on the fixed test split. We report: (i) selector accuracy (fraction of instances where the predicted solver is best), (ii) average rank (lower is better), and for the cross-problem suite also the MiniZinc Borda score. Scripts regenerate all tables/plots from logs.

**Hardware.** Experiments were run on a cluster with nodes equipped with two AMD 7403 processors (24 cores @ 2.8 GHz) and 32 GB RAM per core. We provide the job scripts used on our scheduler; the pipeline also runs on a single workstation (with longer runtimes) using the provided Docker image.

**Determinism and seeds.** Where components are nondeterministic, we set seeds (42) and document any remaining sources of variability (e.g., parallel tree building, AutoSklearn's ensembling). All reported numbers are from a single, fixed split; we include scripts to rerun with new splits and to aggregate across folds if desired.

**Availability.** The anonymized reproduction package includes: code, configs, prompts, generated extractors, and figure/table notebooks.

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

# A APPENDIX A

## A.1 GENERAL PYTHON SCRIPT TEMPLATE GUIDED BY SCRIPT-SYSTEM-PROMPT.MD

```python
# Required script structure
import necessary_modules
# Constants and configuration
CONSTANTS = values
# Function definitions
def helper_functions():
    pass
# Main execution logic
if __name__ == "__main__":
    # Processing logic here
    result_dict = {"key": "value", "results": data}
# MANDATORY: Output results
output_results(result_dict)  # Must be final line
```

Listing 1: General Script Template Structure

## A.2 MINIZINC INSTANCE FEATURE EXTRACTION TEMPLATE GUIDED BY MZN-TUNING-PROMPT.MD

```python
# MANDATORY imports (exact format required)
from lmtune_helpers import input_data, output_results
import networkx as nx
import numpy as np
def main():
    # Get instance data (no file I/O allowed)
    instance_data = input_data()
    # Initialize standardized results structure
    results = {
        "README": "~200 word methodology description",
        "characteristic_1": 0.0,  # Problem size metrics
        "characteristic_2": 0.0,  # Graph properties
        # ... (extract exactly 50 characteristics)
        "characteristic_50": 0.0  # Structural complexity
    }
    # Analyze constraint optimization instance
    # Extract solver-relevant characteristics:
    # - Problem size (variables, constraints)
    # - Graph properties (density, clustering, centrality)
    # - Data distribution (statistical properties)
    # - Structural complexity (symmetries, sparsity)
    # MANDATORY: Return standardized results
    output_results(results)
if __name__ == "__main__":
    main()
```

Listing 2: MiniZinc Instance Feature Extraction Template

## A.3 PARAMETER SETTINGS FOR DIFFERENT ALGORITHM SELECTORS

This section provides details on parameter settings for various algorithm selectors. The following setting has been applied to both random forest training and LLAMMA with the random forest classification mode.

| Parameter | Value | Discription |
|---|---|---|
| n_estimators | 300 | More trees for better performance on complex constraint patterns |
| max_depth | 20 | Deeper trees to capture complex constraint programming relationships |
| min_samples_split | 5 | |
| min_samples_leaf | 2 | |
| resampling_strategy | 'cv' | |
| resampling_strategy_arguments | {'folds': 5} | |
| max_features | 'sqrt' | Standard dimensionality reduction for tree diversity |
| class_weight | 'balanced' | Handle solver class imbalance in dataset |
| random_state | 42 | |

Table 2: Random Forest Hyperparameters

| Parameter | Value |
|---|---|
| time_left_for_this_task | User-specified (300-3600s) |
| per_run_time_limit | time_budget // 30 s |
| initial_configurations_via_metalearning | 25 |
| ensemble_size | 50 |
| resampling_strategy | 'cv' |
| resampling_strategy_arguments | {'folds': 5} |
| scoring_functions | [accuracy/ranking/Borda] |
| memory_limit | 3072 MB |
| tmp_folder | Auto-generated |
| delete_tmp_folder_after_terminate | False |
| random_state | 42 |

Table 3: AutoSklearn Standard Configuration

## A.4 EXPERIMENTAL RESULTS ON FEATURE ANALYSIS

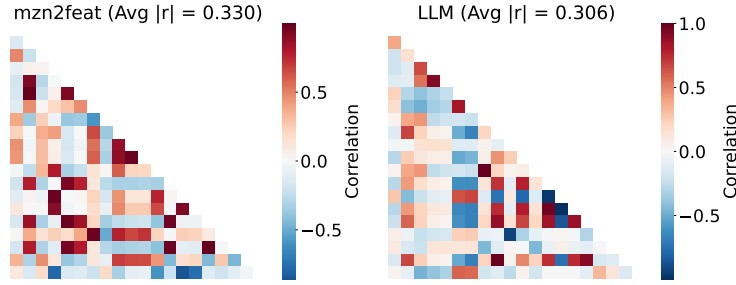

Figure 7: Feature correlation analysis for FLECC problem. LLM features show lower average correlation ($|r| = 0.306$) compared to *mzn2feat* ($|r| = 0.330$), indicating more diverse features.

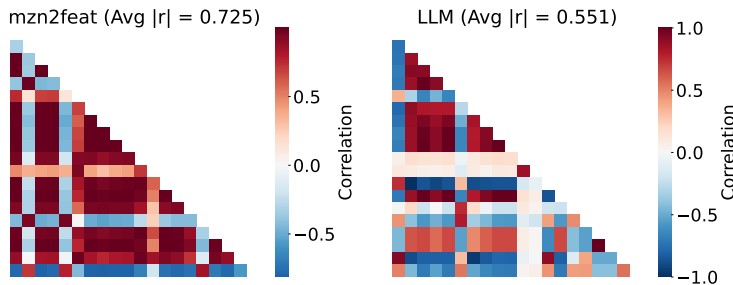

Figure 8: Feature correlation analysis for CS problem. LLM features show significantly lower average correlation ($|r| = 0.551$) compared to *mzn2feat* ($|r| = 0.725$), demonstrating 24% improvement in feature diversity.

### A.5  CORRELATION ANALYSIS ON PROBLEM-GENERIC EXTRACTORS AND *mzn2feat*

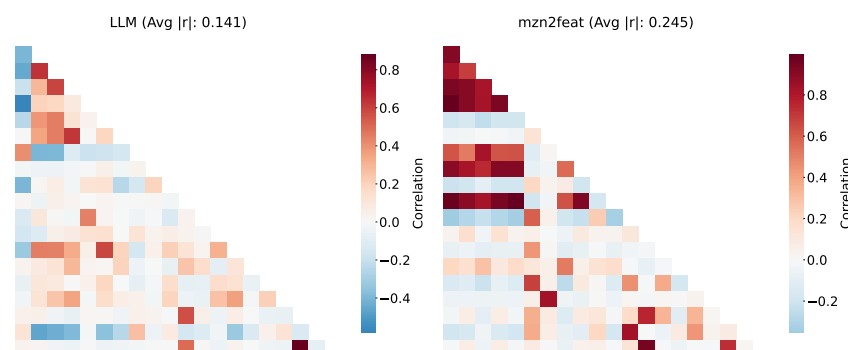

Figure 9: Feature correlation analysis for multiple problems. LLM features show significantly lower average correlation ($|r| = 0.141$) compared to *mzn2feat* ($|r| = 0.245$), demonstrating the improvement in feature diversity.

### A.6  TOP FEATURES DISTRIBUTION ANALYSIS ON PROBLEM-GENERIC EXTRACTORS AND *mzn2feat*

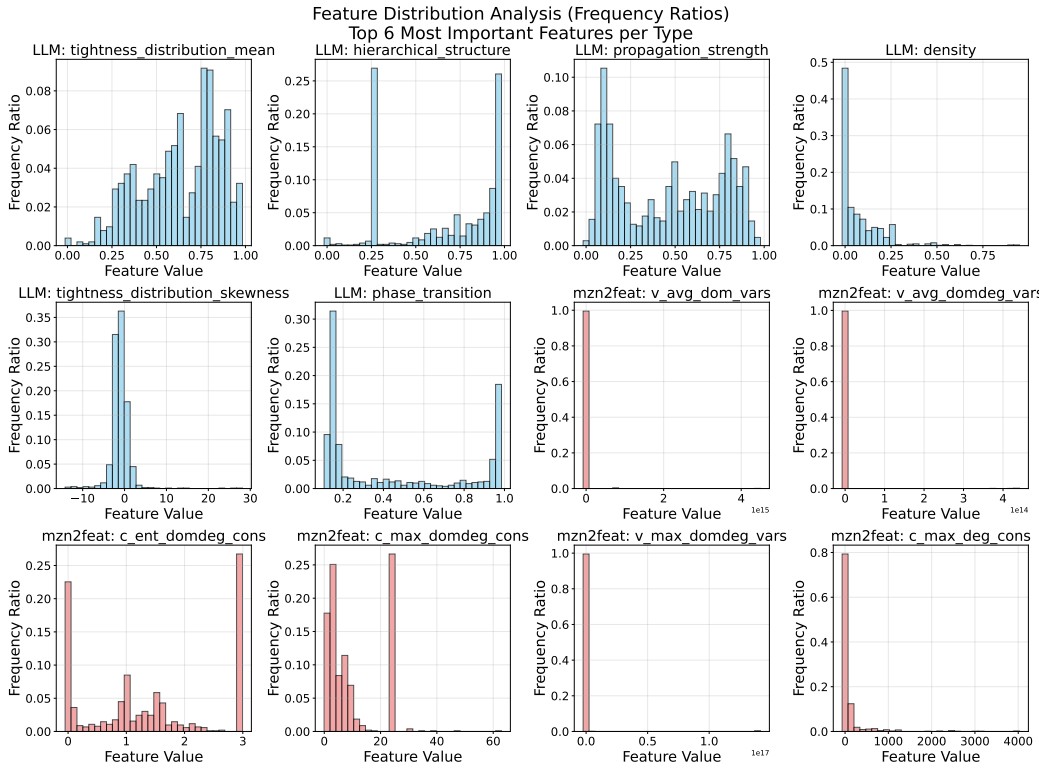

Figure 10: For LLM-based generic features and *mzn2feat*-based features, we compare the instance distribution on different single top features (Top-6 features). We can see that for LLM-based features, the instances are distributed more evenly across different feature values, which potentially provides more useful information to the algorithm selectors for classification.

## A.7 EXPERIMENTAL RESULTS OF PROBLEM-GENERIC FEATURE EXTRACTOR AND *mzn2feat*-BASED ALGORITHM SELECTORS ON TRAINING SETS WITH $Acc$ AS LOSS FUNCTIONS.

Here we also show the results of *mzn2feat* and *LLM2feat*-based algorithm selectors' performance on training sets.

Loss function=$Acc$

| Training | VRP | | CS | | FLECC | | Suite (200+Problems) | | |
|---|---|---|---|---|---|---|---|---|---|
| | Acc | Rank | Acc | Rank | Acc | Rank | Acc | Rank | Borda |
| SB | 78.7% | 1.237 | 49.4% | 1.601 | 80.6% | 1.377 | 27.0% | 2.667 | 0.694 |
| *mzn2feat*+AF | 78.7% | 1.237 | 49.4% | 1.601 | 80.6% | 1.377 | 27.0% | 2.667 | 0.694 |
| *mzn2feat*+RF | 91.7% | 1.091 | 83.5% | 1.231 | 88.3% | 1.232 | 81.0% | 1.186 | 1.491 |
| *mzn2feat*+LLAMMA | 92.0% | 1.089 | 87.5% | 1.248 | 91.5% | 1.167 | × | × | × |
| *mzn2feat*+AutoSK | 87.2% | 1.138 | 65.0% | 1.723 | 85.0% | 1.296 | 64.3% | 1.322 | 1.402 |
| *LLM2feat*+RF | 97.1% | 1.032 | 87.9% | 1.164 | 95.9% | 1.083 | 97.9% | 1.016 | 1.653 |
| *LLM2feat*+LLAMMA | 97.8% | 1.025 | 89.9% | 1.164 | 98.3% | 1.039 | × | × | × |
| *LLM2feat*+AutoSK | 87.8% | 1.133 | 68.9% | 1.618 | 95.1% | 1.103 | 89.8% | 1.078 | 1.608 |

Table 4: Experimental results of problem-specific and problem-generic feature extractors and *mzn2feat*-based algorithm selectors on training sets with $Acc$ as loss functions.

## A.8 EXPERIMENTAL RESULTS OF PROBLEM-SPECIFIC FEATURE EXTRACTOR AND *mzn2feat*-BASED ALGORITHM SELECTORS ON TEST SETS WITH $Rank$ AS LOSS FUNCTIONS.

Here, we also present the results of the *mzn2feat* and *LLM2feat*-based algorithm selectors' performance on testing sets using $Rank$ as the loss function.

Loss function=$Rank$

| (Testing) | VRP | | CS | | FLECC | |
|---|---|---|---|---|---|---|
| | Acc | Rank | Acc | Rank | Acc | Rank |
| SB | 79.5% | 1.221 | 49.0% | 1.616 | 78.2% | 1.420 |
| *mzn2feat*+RF | 83.0% | 1.184 | 58.5% | 1.680 | 79.5% | 1.391 |
| *mzn2feat*+AutoSK | 84.4% | 1.166 | 62.1% | 1.787 | 79.4% | 1.394 |
| *LLM2feat*+RF | 85.7% | 1.155 | 61.3% | 1.681 | 83.5% | 1.338 |
| *LLM2feat*+AutoSK | 85.4% | 1.160 | 64.0% | 1.738 | 84.6% | 1.310 |

Table 5: Experimental results of problem-specific feature extractor and *mzn2feat*-based algorithm selectors on test sets with $Rank$ as loss functions.

## A.9 EXPERIMENTAL RESULTS OF PROBLEM-SPECIFIC FEATURE EXTRACTOR AND *mzn2feat*-BASED ALGORITHM SELECTORS ON TRAINING SETS WITH $Rank$ AS LOSS FUNCTIONS.

Here, we also present the results of the *mzn2feat* and *LLM2feat*-based algorithm selectors' performance on training sets with $Rank$ as the loss function.

## A.10 LLM MODEL SELECTION AND SENSITIVITY ANALYSIS OF *LLM2feat* EXTRACTORS

We evaluate several LLMs as the backend of our agent. Current results indicate that strong agentic models (o4-mini, Claude Sonnet) reliably follow the check-fix-verify protocol, while smaller open-source models (Llama 3.3, DeepSeek R1) fail to do so under our prompts (see Cost Analysis Appendix). To analyze the sensitivity of the framework to LLM-specific variations, we select OpenAI o4-mini-2025-04-16 for the problem-specific framework and generate 10 different feature extractors for each problem. We use the most classical tool chain, RF, and the most advanced tool chain, AutoSK, for algorithm selector training. As shown in the following tables, each table lists the results of three different LLM-based feature extractors (LLM-timestamp) for the same training setting. For

Loss function=$Rank$

| Training | VRP | | CS | | FLECC | |
|---|---|---|---|---|---|---|
| | Acc | Rank | Acc | Rank | Acc | Rank |
| SB | 78.7% | 1.237 | 49.4% | 1.601 | 80.6% | 1.377 |
| *mzn2feat*+RF | 91.7% | 1.091 | 83.5% | 1.231 | 88.3% | 1.232 |
| *mzn2feat*+AutoSK | 87.2% | 1.138 | 65.0% | 1.723 | 85.0% | 1.296 |
| *LLM2feat*+RF | 97.1% | 1.032 | 87.9% | 1.164 | 95.9% | 1.083 |
| *LLM2feat*+AutoSK | 87.8% | 1.133 | 68.9% | 1.618 | 95.1% | 1.103 |

Table 6: Experimental results of problem-specific feature extractor and *mzn2feat*-based algorithm selectors on training sets with $Rank$ as loss functions.

the full results of all 10 LLM-based feature extractors, we have them in the Appendix C. The accuracy and the average ranking of their algorithm selectors fluctuate in a quite small range (usually less than 2%). Meanwhile, we also compare the $Acc$ and $Rank$ obtained by different algorithm selectors based on *mzn2feat* and all three *LLM2feat* extractors. We can see that all *LLM2feat*-based algorithm selectors consistently outperform *mzn2feat*-based algorithm selectors on various metrics ($Acc$ and $Rank$).

Regarding the selection of LLM models in the problem-generic framework, we use Anthropic claude-sonnet-4-20250514. Here, we have to deal with 227 problems related to extractor generation, which is conveniently tackled with the subagent functionality of Claude.

Table 7: *LLM2feat*+RF Performance for FLECC Problem

| Feature Extractor | Loss Function | Train Acc | Test Acc | Train Rank | Test Rank |
|---|---|---|---|---|---|
| **Single Best (gecode)** | | 0.806 | 0.782 | 1.377 | 1.420 |
| *mzn2feat* | accuracy | 0.959 | 0.764 | 1.063 | 1.426 |
| *mzn2feat* | ranking | 0.883 | 0.795 | 1.232 | 1.391 |
| LLM-20250908123730 | accuracy | 0.981 | 0.847 | 1.029 | 1.286 |
| LLM-20250908123925 | accuracy | 0.996 | 0.818 | 1.008 | 1.353 |
| LLM-20250908124149 | accuracy | 0.994 | 0.836 | 1.011 | 1.315 |
| LLM-20250908123730 | ranking | 0.959 | 0.835 | 1.083 | 1.338 |
| LLM-20250908123925 | ranking | 0.935 | 0.809 | 1.121 | 1.384 |
| LLM-20250908124149 | ranking | 0.952 | 0.826 | 1.095 | 1.358 |

Table 8: *LLM2feat*+RF Performance for CS Problem

| Feature Extractor | Loss Function | Train Acc | Test Acc | Train Rank | Test Rank |
|---|---|---|---|---|---|
| **Single Best (cplex)** | | 0.494 | 0.490 | 1.601 | 1.616 |
| *mzn2feat* | accuracy | 0.857 | 0.547 | 1.339 | 1.943 |
| *mzn2feat* | ranking | 0.835 | 0.585 | 1.231 | 1.680 |
| LLM-20250908123608 | accuracy | 0.903 | 0.577 | 1.193 | 1.874 |
| LLM-20250908123905 | accuracy | 0.913 | 0.578 | 1.181 | 1.861 |
| LLM-20250908124041 | accuracy | 0.906 | 0.581 | 1.202 | 1.862 |
| LLM-20250908123608 | ranking | 0.868 | 0.607 | 1.179 | 1.700 |
| LLM-20250908123905 | ranking | 0.879 | 0.613 | 1.165 | 1.681 |
| LLM-20250908124041 | ranking | 0.875 | 0.606 | 1.171 | 1.704 |

Table 9: *LLM2feat*+RF Performance for VRP Problem

| Feature Extractor | Loss Function | Train Acc | Test Acc | Train Rank | Test Rank |
|---|---|---|---|---|---|
| **Single Best (scip)** | | 0.787 | 0.795 | 1.237 | 1.221 |
| *mzn2feat* | accuracy | 0.947 | 0.824 | 1.057 | 1.195 |
| *mzn2feat* | ranking | 0.916 | 0.830 | 1.091 | 1.184 |
| LLM-20250908115627 | accuracy | 0.993 | 0.850 | 1.008 | 1.169 |
| LLM-20250908121942 | accuracy | 0.994 | 0.848 | 1.007 | 1.170 |
| LLM-20250908123205 | accuracy | 0.985 | 0.853 | 1.016 | 1.165 |
| LLM-20250908115627 | ranking | 0.971 | 0.852 | 1.032 | 1.163 |
| LLM-20250908121942 | ranking | 0.972 | 0.857 | 1.032 | 1.155 |
| LLM-20250908123205 | ranking | 0.970 | 0.852 | 1.034 | 1.163 |

Table 10: *LLM2feat*+AutoSK Performance for FLECC Problem

| Feature Extractor | Loss Function | Train Acc | Test Acc | Train Rank | Test Rank |
|---|---|---|---|---|---|
| **Single Best (gecode)** | | 0.806 | 0.782 | 1.377 | 1.420 |
| *mzn2feat* | accuracy | 0.850 | 0.792 | 1.295 | 1.396 |
| *mzn2feat* | ranking | 0.850 | 0.794 | 1.296 | 1.394 |
| LLM-20250908124149 | accuracy | 0.952 | 0.844 | 1.101 | 1.308 |
| LLM-20250908123730 | accuracy | 0.951 | 0.846 | 1.104 | 1.310 |
| LLM-20250908123925 | accuracy | 0.908 | 0.831 | 1.187 | 1.342 |
| LLM-20250908124149 | ranking | 0.951 | 0.845 | 1.103 | 1.308 |
| LLM-20250908123730 | ranking | 0.951 | 0.846 | 1.104 | 1.310 |
| LLM-20250908123925 | ranking | 0.911 | 0.836 | 1.183 | 1.322 |

Table 11: *LLM2feat*+AutoSK Performance for CS Problem

| Feature Extractor | Loss Function | Train Acc | Test Acc | Train Rank | Test Rank |
|---|---|---|---|---|---|
| **Single Best (cplex)** | | 0.494 | 0.490 | 1.601 | 1.616 |
| *mzn2feat* | accuracy | 0.650 | 0.621 | 1.723 | 1.787 |
| *mzn2feat* | ranking | 0.650 | 0.621 | 1.723 | 1.787 |
| LLM-20250908124041 | accuracy | 0.671 | 0.640 | 1.646 | 1.735 |
| LLM-20250908123905 | accuracy | 0.679 | 0.634 | 1.667 | 1.781 |
| LLM-20250908123608 | accuracy | 0.686 | 0.635 | 1.652 | 1.782 |
| LLM-20250908124041 | ranking | 0.670 | 0.640 | 1.655 | 1.738 |
| LLM-20250908123905 | ranking | 0.675 | 0.638 | 1.655 | 1.766 |
| LLM-20250908123608 | ranking | 0.689 | 0.639 | 1.618 | 1.743 |

Table 12: *LLM2feat*+AutoSK Performance for VRP Problem

| Feature Extractor | Loss Function | Train Acc | Test Acc | Train Rank | Test Rank |
|---|---|---|---|---|---|
| **Single Best (scip)** | | 0.787 | 0.795 | 1.237 | 1.221 |
| *mzn2feat* | accuracy | 0.872 | 0.844 | 1.138 | 1.166 |
| *mzn2feat* | ranking | 0.872 | 0.844 | 1.138 | 1.166 |
| LLM-20250908121942 | accuracy | 0.851 | 0.852 | 1.161 | 1.162 |
| LLM-20250908123205 | accuracy | 0.880 | 0.857 | 1.131 | 1.157 |
| LLM-20250908115627 | accuracy | 0.872 | 0.854 | 1.138 | 1.160 |
| LLM-20250908121942 | ranking | 0.851 | 0.852 | 1.161 | 1.162 |
| LLM-20250908123205 | ranking | 0.878 | 0.854 | 1.133 | 1.160 |
| LLM-20250908115627 | ranking | 0.859 | 0.850 | 1.152 | 1.162 |

### A.11 Definitions

**Definition 1** *The* Single Best Solver *(SB) is the algorithm $A^{SB} \in \mathcal{P}$ that achieves the best overall performance with respect to the performance metric $PM$ across the entire set of instances $I$, i.e.,*

$$A^{SB} = \arg\max_{A \in \mathcal{P}} PM(A, I).$$

*The Single Best Solver strategy corresponds to selecting the Single Best Solver for all instances.*

**Definition 2** *The* Virtual Best Solver *(VBS) is the (hypothetical) per-instance selector that, for each instance $i \in I$, chooses the algorithm $A \in \mathcal{P}$ that achieves the best performance on that instance with respect to $PM$. Formally, its performance is defined as*

$$PM(VBS, I) = \sum_{i \in I} \max_{A \in \mathcal{P}} PM(A, \{i\}).$$

*The VBS serves as an upper bound on the achievable performance of any AS strategy, assuming perfect knowledge of per-instance performance.*

**Definition 3** *A Large Language Model agent,* **LLM-agent** *(Yao et al., 2023) is a tuple $\mathcal{A} = (L, T, M, \pi, E)$, where $L$ is a large language model used for reasoning and generation, $T$ is a set of external tools or APIs accessible by the agent, $M$ is the memory module (short-term and/or long-term), $\pi$ is the prompting policy that maps observations and history to model inputs, $E$ is the environment with which the agent interacts via observations and actions.*

*The LLM-agent agent operates in a loop: $o_t \xrightarrow{\pi} p_t \xrightarrow{L} a_t \xrightarrow{T,E} o_{t+1}$ where $o_t$ is the observation at time $t$, $p_t$ is the constructed prompt, $a_t$ is the action (e.g., tool call, output), and $o_{t+1}$ is the new observation.*

## B Appendix: Qualitative Feature Analysis

Generally, our LLM-synthesized extractors outperform classical baselines because they (a) preserve high-level structure that is lost in flat encodings, (b) materialize solver-relevant semantics as explicit features, and (c) adapt feature definitions to the problem family (specific) or canonically factor them through a typed variable–constraint graph (generic).

As we analyze in Appendix A.6, we list the top six important features for both *LLM2feat* and *mzn2feat*, where most of the features from *LLM2feat* are structure-related from the generic graph perspective, including hierarchical, density, and skewness, etc.

More concretely, we list and categorize some features from *LLM2feat* into different domains and briefly discuss why these differences result in higher efficiency for the algorithm selection.

- Preserving structural information (vs. *mzn2feat*): *mzn2feat* Amadini et al. (2014) contains mainly flattened features, that is, transforming constraint modelling language (minizinc) into a long list or table of primitive constraints with many structures removed. However, richer structures can dominate over the cheap, flattened features from some previous studies (Dalla et al., 2023; Shavit & Hoos, 2024). Our *LLM2feat* constructs a typed bipartite graph that contains variables and constraints, thereby preserving the incidence structure, arity, and heterogeneity. This enables statistics that matter for solver behaviors: degree/weight dispersion and skew (heterogeneous coupling hints branching difficulty); Constraint overlap motifs (shared-variable cliques indicate tight substructures), and so on.

- Semantic and solver-aware quantities that flat features miss: the LLM reads the models (.dzn files) (objective, global constraints, parameter roles), generated code computes proxy quantities solvers implicitly exploit tightness, propagation strength that approximate domain reduction ratios and supports per constraint family (e.g., alldifferent, table), and their distributional summaries (mean/variance/skew).

- Problem-aware feature crafting (specialized extractor): When the agent sees a specific family, it instantiates domain priors as interpretable features. For example, in VRP, demand concentration, depot eccentricity, and route-length lower bounds (such as minimum spanning tree surrogates) can be used to separate MIP (Gurobi, CPLEX, SCIP) and CP (OR-Tool, Gecode) by leveraging the solvers' strengths (Moreno-Scott et al., 2016).

## C  APPENDIX: TRANSFORMER COMPARISON

Compared with the expert-based feature set (mzn2feat), as shown in the paper, we obtain better results across various metrics (accuracy, ranking, and scores) for problem-specific and problem-generic frameworks. Furthermore, in a more detailed manner, we present the head-to-head accuracy and ranking comparison between the problem-specific framework and transformer-based feature extractors (Pellegrino et al., 2025).

From the official repository [5] we obtain 20 feature sets (because the authors built different variants of the neural networks (Pellegrino et al., 2025)). We use the same toolchains (Random Forest and AutoSklearn) to train algorithm selectors based on different features for two FLECC and car sequencing problems. We could only compare these two problems because they are also used in their paper, and we can directly use the features they produced for a fair comparison. We find that our LLM2feat-based algorithm selectors (from 10 different runs) consistently achieve better accuracy and ranking than selectors based on other features, including transformer-based and mzn2feat-based methods. From Table 13, Table 15, Table 14 and Table 16, even the worst run of our framework can mostly get higher accuracy and ranking than the best variants of the transformer-based method.

Considering the feature extraction time consumption, all transformer-based features of an instance are computed within 1 second on a GPU-enabled machine, as reported in the original paper. For *mzn2feat* and *LLM2feat*, we set timeouts of 3 minutes and 1 minute, respectively, for CPU wall time. Regarding the whole optimization process, all these timeouts are intangible.

## D  APPENDIX: COST ANALYSIS

To systematically understand the efficiency of LLM generation, we run *LLM2feat* on OpenAI o4-mini-2025-04-16, Anthropic claude-sonnet-4-20250514, DeepSeek R1, and llama 3.3-8b-instruct without a timeout. As we explain the experiential setting, we set a timeout for real generation of feature extractors in case the LLM takes too long, which would result in excessive cost as the context length increases with iterations.

We experiment with ChatGPT O4-mini and Claude Sonnet 4; they consistently generate extractors (10 times out of 10 trials). However, for DeepSeek R1 and llama 3.3-8b-instruct, we find that they fail to generate extractors all 10 times because they could not follow the defined procedures in the prompt (insert-check/fix-verify).

Consequently, Table 17 presents statistical results from 30 runs of ChatGPT O4-mini on different problems, without timeouts, until extractors are generated. Once *LLM2feat* produces a feature extractor, it can be used to extract features an unlimited number of times for different instances. Therefore, the cost (less than 0.3 dollars) is reasonably practical in real-life application problems. The one-time extractor generation cost ($\approx$210 s, \$0.27) is negligible compared to running each solver for up to 20 minutes per instance.

---

[5] https://github.com/SeppiaBrilla/EFE_project/tree/master/data/features

Table 13: Car Sequencing: Random Forest with Accuracy Loss

| Type | Extractor | Features | Accuracy | | Avg Rank | |
|---|---|---|---|---|---|---|
| | | | Train | Test | Train | Test |
| LLM2feat | LLM2feat-1 | 50 | 0.909 | 0.583 | 1.187 | 1.857 |
| | LLM2feat-2 | 50 | 0.906 | 0.581 | 1.202 | 1.862 |
| | LLM2feat-3 | 50 | 0.920 | 0.579 | 1.170 | 1.855 |
| | LLM2feat-4 | 50 | 0.908 | 0.579 | 1.188 | 1.867 |
| | LLM2feat-5 | 50 | 0.913 | 0.578 | 1.181 | 1.861 |
| | LLM2feat-6 | 50 | 0.899 | 0.578 | 1.213 | 1.867 |
| | LLM2feat-7 | 50 | 0.903 | 0.577 | 1.193 | 1.874 |
| | LLM2feat-8 | 50 | 0.915 | 0.577 | 1.180 | 1.872 |
| | LLM2feat-9 | 50 | 0.899 | 0.575 | 1.197 | 1.876 |
| | LLM2feat-10 | 50 | 0.904 | 0.573 | 1.193 | 1.875 |
| mzn2feat | mzn2feat | 95 | 0.857 | 0.547 | 1.339 | 1.943 |
| trans2feat | com-6 | 116 | 0.869 | 0.529 | 1.344 | 2.004 |
| | com-9 | 116 | 0.833 | 0.529 | 1.425 | 1.982 |
| | com-3 | 116 | 0.861 | 0.521 | 1.384 | 2.030 |
| | com-2 | 116 | 0.850 | 0.518 | 1.398 | 2.039 |
| | com-1 | 116 | 0.840 | 0.512 | 1.422 | 2.048 |
| | com-5 | 116 | 0.838 | 0.508 | 1.420 | 2.044 |
| | com-0 | 116 | 0.857 | 0.502 | 1.391 | 2.081 |
| | com-7 | 116 | 0.843 | 0.502 | 1.415 | 2.070 |
| | com-8 | 116 | 0.773 | 0.501 | 1.569 | 2.083 |
| | com-4 | 116 | 0.826 | 0.496 | 1.438 | 2.088 |
| | neural-7 | 116 | 0.789 | 0.482 | 1.522 | 2.110 |
| | neural-6 | 116 | 0.558 | 0.418 | 2.394 | 2.637 |
| | neural-0 | 116 | 0.557 | 0.411 | 2.371 | 2.627 |
| | neural-4 | 116 | 0.472 | 0.383 | 2.601 | 2.758 |
| | neural-8 | 116 | 0.428 | 0.368 | 2.723 | 2.836 |
| | neural-2 | 116 | 0.343 | 0.333 | 2.861 | 2.886 |
| | neural-9 | 116 | 0.366 | 0.326 | 2.735 | 2.828 |
| | neural-3 | 116 | 0.329 | 0.315 | 2.975 | 3.026 |
| | neural-5 | 116 | 0.309 | 0.314 | 2.978 | 2.985 |
| | neural-1 | 116 | 0.319 | 0.310 | 3.009 | 3.039 |

Table 14: FLECC: Random Forest with Accuracy Loss

| Type | Extractor | Features | Accuracy | | Avg Rank | |
|---|---|---|---|---|---|---|
| | | | Train | Test | Train | Test |
| LLM2feat | LLM2feat-1 | 50 | 0.981 | 0.847 | 1.029 | 1.286 |
| | LLM2feat-2 | 50 | 0.989 | 0.842 | 1.018 | 1.310 |
| | LLM2feat-3 | 50 | 0.984 | 0.842 | 1.025 | 1.305 |
| | LLM2feat-4 | 50 | 0.994 | 0.836 | 1.011 | 1.315 |
| | LLM2feat-5 | 50 | 0.995 | 0.835 | 1.009 | 1.325 |
| | LLM2feat-6 | 50 | 0.980 | 0.831 | 1.033 | 1.324 |
| | LLM2feat-7 | 50 | 0.989 | 0.826 | 1.019 | 1.334 |
| | LLM2feat-8 | 50 | 0.996 | 0.818 | 1.008 | 1.353 |
| | LLM2feat-9 | 50 | 0.994 | 0.818 | 1.012 | 1.360 |
| | LLM2feat-10 | 50 | 0.976 | 0.816 | 1.041 | 1.350 |
| mzn2feat | mzn2feat | 95 | 0.959 | 0.764 | 1.063 | 1.426 |
| trans2feat | com-3 | 116 | 0.803 | 0.773 | 1.389 | 1.447 |
| | com-2 | 116 | 0.800 | 0.772 | 1.399 | 1.459 |
| | com-1 | 116 | 0.765 | 0.746 | 1.532 | 1.552 |
| | neural-9 | 116 | 0.958 | 0.745 | 1.067 | 1.480 |
| | neural-7 | 116 | 0.958 | 0.743 | 1.072 | 1.490 |
| | com-8 | 116 | 0.941 | 0.734 | 1.107 | 1.517 |
| | com-7 | 116 | 0.949 | 0.732 | 1.085 | 1.501 |
| | com-9 | 116 | 0.933 | 0.731 | 1.108 | 1.510 |
| | neural-8 | 116 | 0.947 | 0.727 | 1.088 | 1.522 |
| | com-4 | 116 | 0.727 | 0.708 | 1.637 | 1.660 |
| | com-5 | 116 | 0.688 | 0.673 | 1.775 | 1.785 |
| | neural-1 | 116 | 0.688 | 0.669 | 1.805 | 1.825 |
| | neural-0 | 116 | 0.612 | 0.590 | 2.103 | 2.136 |
| | com-6 | 116 | 0.472 | 0.444 | 2.172 | 2.267 |
| | neural-6 | 116 | 0.095 | 0.114 | 2.466 | 2.455 |
| | neural-3 | 116 | 0.095 | 0.111 | 2.510 | 2.496 |
| | neural-2 | 116 | 0.086 | 0.108 | 2.610 | 2.573 |
| | com-0 | 116 | 0.030 | 0.033 | 3.933 | 3.968 |
| | neural-5 | 116 | 0.027 | 0.031 | 3.957 | 3.985 |
| | neural-4 | 116 | 0.027 | 0.031 | 3.957 | 3.985 |

Table 15: Car Sequencing: Autosklearn with Accuracy Loss. TO: Timeout means failure of training due to a long timeout (1800 seconds). Because Autosklearn integrates tuning techniques and ensemble learning methods, it sometimes requires a considerable amount of time (Feurer et al., 2020). For fair comparisons, we ignore these runs.

| Type | Extractor | Features | Accuracy | | Avg Rank | |
|---|---|---|---|---|---|---|
| | | | Train | Test | Train | Test |
| LLM2feat | LLM2feat-1 | 50 | 0.671 | 0.640 | 1.646 | 1.735 |
| | LLM2feat-2 | 50 | 0.686 | 0.635 | 1.652 | 1.782 |
| | LLM2feat-3 | 50 | 0.679 | 0.634 | 1.667 | 1.781 |
| | LLM2feat-4 | 50 | 0.709 | 0.630 | 1.578 | 1.756 |
| | LLM2feat-5 | 50 | 0.669 | 0.629 | 1.631 | 1.729 |
| | LLM2feat-6 | 50 | 0.641 | 0.616 | 1.707 | 1.788 |
| | LLM2feat-7 | 50 | 0.651 | 0.607 | 1.696 | 1.802 |
| | LLM2feat-8 | 50 | 0.660 | 0.529 | 1.738 | 2.011 |
| | LLM2feat-9 | 50 | 0.658 | 0.528 | 1.754 | 2.017 |
| | LLM2feat-10 (TO) | 50 | | | | ... |
| mzn2feat | mzn2feat | 95 | 0.650 | 0.621 | 1.723 | 1.787 |
| trans2feat | com-4 | 116 | 0.867 | 0.545 | 1.271 | 1.931 |
| | com-7 | 116 | 0.869 | 0.542 | 1.270 | 1.927 |
| | neural-9 | 116 | 0.568 | 0.512 | 2.142 | 2.248 |
| | neural-8 | 116 | 0.521 | 0.499 | 2.435 | 2.492 |
| | neural-1 | 116 | 0.521 | 0.498 | 2.429 | 2.493 |
| | neural-6 | 116 | 0.518 | 0.495 | 2.420 | 2.482 |
| | neural-3 | 116 | 0.519 | 0.492 | 2.453 | 2.522 |
| | neural-5 | 116 | 0.511 | 0.491 | 2.493 | 2.553 |
| | neural-2 | 116 | 0.534 | 0.491 | 2.405 | 2.500 |
| | com-6 | 116 | 0.582 | 0.483 | 1.971 | 2.164 |
| | com-9 | 116 | 0.570 | 0.468 | 1.998 | 2.183 |
| | neural-4 | 116 | 0.558 | 0.465 | 2.419 | 2.570 |
| | com-2 | 116 | 0.570 | 0.462 | 2.015 | 2.224 |
| | com-5 | 116 | 0.545 | 0.456 | 2.063 | 2.236 |
| | com-8 | 116 | 0.523 | 0.452 | 2.133 | 2.270 |
| | neural-7 | 116 | 0.535 | 0.445 | 2.088 | 2.256 |
| | neural-0 | 116 | 0.397 | 0.358 | 2.709 | 2.798 |
| | com-1 (TO) | 116 | | | | ... |

Table 16: FLECC: Autosklearn with Accuracy Loss. TO: Timeout means failure of training due to a long timeout (1800 seconds). Because Autosklearn integrates tuning techniques and ensemble learning methods, it may sometimes require a considerable amount of time (Feurer et al., 2020). For fair comparisons, we ignore these runs.

| Type | Extractor | Features | Accuracy | | Avg Rank | |
|---|---|---|---|---|---|---|
| | | | Train | Test | Train | Test |
| LLM2feat | LLM2feat-1 | 50 | 0.951 | 0.846 | 1.104 | 1.310 |
| | LLM2feat-2 | 50 | 0.946 | 0.845 | 1.113 | 1.311 |
| | LLM2feat-3 | 50 | 0.952 | 0.844 | 1.101 | 1.308 |
| | LLM2feat-4 | 50 | 0.886 | 0.838 | 1.231 | 1.325 |
| | LLM2feat-5 | 50 | 0.924 | 0.837 | 1.157 | 1.324 |
| | LLM2feat-6 | 50 | 0.904 | 0.836 | 1.195 | 1.330 |
| | LLM2feat-7 | 50 | 0.908 | 0.831 | 1.187 | 1.342 |
| | LLM2feat-8 | 50 | 0.897 | 0.830 | 1.209 | 1.342 |
| | LLM2feat-9 | 50 | 0.872 | 0.828 | 1.250 | 1.340 |
| | LLM2feat-10 | 50 | 0.919 | 0.822 | 1.164 | 1.356 |
| mzn2feat | mzn2feat | 95 | 0.850 | 0.792 | 1.295 | 1.396 |
| trans2feat | com-7 | 116 | 0.819 | 0.784 | 1.357 | 1.417 |
| | com-8 | 116 | 0.809 | 0.783 | 1.371 | 1.418 |
| | neural-9 | 116 | 0.816 | 0.783 | 1.361 | 1.417 |
| | com-4 | 116 | 0.807 | 0.783 | 1.375 | 1.419 |
| | neural-8 | 116 | 0.826 | 0.783 | 1.345 | 1.417 |
| | com-2 | 116 | 0.806 | 0.782 | 1.377 | 1.420 |
| | neural-1 | 116 | 0.806 | 0.782 | 1.377 | 1.420 |
| | neural-2 | 116 | 0.806 | 0.782 | 1.377 | 1.420 |
| | neural-6 | 116 | 0.806 | 0.782 | 1.377 | 1.420 |
| | com-6 | 116 | 0.807 | 0.782 | 1.374 | 1.420 |
| | com-9 | 116 | 0.807 | 0.781 | 1.374 | 1.422 |
| | neural-7 | 116 | 0.806 | 0.781 | 1.376 | 1.422 |
| | neural-4 | 116 | 0.082 | 0.083 | 2.830 | 2.817 |
| | neural-5 | 116 | 0.082 | 0.083 | 2.830 | 2.817 |
| | neural-3 (TO) | | | | | ... |

Table 17: Statistics of 30 Runs of LLM2feat

| Metric | Value |
|---|---|
| Avg Iterations | 17.3 |
| Avg Wall Time | 210.0 s |
| Avg Input Tokens | 259913 |
| Avg Output Tokens | 19238 |
| Avg Total Tokens | 279150 |
| Avg Cost | $0.2695 |

