# OpenReview forum: "Synthesizing Feature Extractors: An Agentic Approach for Algorithm Selection"
_ICLR.cc/2026/Conference — Submitted to ICLR 2026_

### Official Review · Reviewer_XnMH · 2025-10-29

**Soundness:** 2
**Presentation:** 1
**Contribution:** 2
**Rating:** 4
**Confidence:** 3

**Summary:**

This paper tackles automated feature engineering for algorithm selection in constraint optimization. Instead of hand-crafted features or neural embeddings, the authors propose using LLMs to generate executable Python scripts that serve as interpretable feature extractors. The scripts take MiniZinc problem descriptions as input and produce graph-based structural features for training algorithm selectors.

Two frameworks are presented: problem-specific (tailored extractors for individual problem types) and problem-generic (universal extractor using standardized bipartite graph representation). Both operate through an agentic LLM workflow that generates, validates, and corrects Python code.

The authors evaluate on algorithm selection across 227 combinatorial problems from MiniZinc Challenge benchmarks using a portfolio of 5 solvers (Gurobi, CPLEX, SCIP, Gecode, OR-Tools). Their LLM-generated features (LLM2feat) achieve 58.8% accuracy versus 48.6% for the baseline human-engineered extractor (mzn2feat). The features show lower correlation, better utilization efficiency, and consistent improvements across different ML models.

The core claim is that LLMs can automatically generate feature extractors that outperform human-engineered ones while maintaining interpretability, democratizing access to algorithm selection techniques.

**Strengths:**

**Novel application to constraint programming:** The application of LLM-based feature extractor synthesis to the algorithm selection domain for constraint programming is interesting. The two-level approach (LLM generates code that builds graphs, then extracts features) and the problem-generic framework using universal bipartite graph representation appear to be new contributions to this specific domain.

**Comprehensive evaluation for problem-generic framework:** The problem-generic framework is tested across 227 problems spanning 17 years of MiniZinc benchmarks, which is quite thorough. The comparison includes multiple ML models (RF, AutoSklearn, AutoFolio, LLAMMA) and multiple metrics (accuracy, ranking, Borda scores). However, the problem-specific framework is only evaluated on 3 problem types (VRP, CS, FLECC).

**Consistent improvements:** LLM2feat features outperform mzn2feat [1] across all tested scenarios. The 10+ percentage point accuracy improvements are substantial and hold across different problem types.

**Feature quality analysis:** The correlation analysis, utilization efficiency metrics, and feature importance distributions provide good evidence that LLM-generated features are genuinely better, not just different.

**Practical impact:** The approach could enable algorithm selection for new problem domains where manual feature engineering is prohibitive.

**Weaknesses:**

**Limited baseline comparisons:** The evaluations only compare against a single, non-neural baseline (mzn2feat [1]). What about recent deep learning approaches mentioned in the intro (Pellegrino et al. 2025 [4], Zhang et al. 2024 [5])? The claim that interpretable features are essential needs stronger empirical justification against end-to-end learned representations.

**Venue fit concerns:** Given the specialized focus on constraint programming and algorithm selection for combinatorial optimization, this work might be better suited for a venue like the International Conference on Principles and Practice of Constraint Programming (CP) where the audience would have more domain context and the contributions would be more directly appreciated. For a general AI audience at ICLR, the paper needs substantially more background and motivation.

**Missing related work section:** The paper lacks a proper related work section that would help readers unfamiliar with this area understand prior work on automated feature extraction for CSPs and algorithm selection. Without this context, it's difficult to assess the significance of the stated contributions. The brief mentions in the intro are insufficient.

**Overstated novelty claim:** The authors claim "to the best of our knowledge, we have introduced the first framework for the automatic construction of feature extractors for combinatorial problems." However, a cursory search suggests work exists on LLM-based automated feature engineering through program synthesis:
  CAAFE (Hollmann et al. 2024) [2] uses LLMs to generate synthetic features through code generation with interpretable, human-in-the-loop approach
  AS-LLM (2024) [3] uses LLMs to automatically extract algorithm representations from code for algorithm selection
  Broader surveys on LLMs for ML workflows [6] document extensive work on automated feature engineering via LLM code generation

  What appears actually novel is the specific application to CP/algorithm selection with the two-level architecture (LLM→code→graph→features) and the universal bipartite graph representation for the generic framework. The paper should revise claims to accurately reflect what's new.

**Limited domain generalization:** All 227 problems in the evaluation come from MiniZinc benchmarks. The generalizability argument would be much stronger if the authors could demonstrate the approach works on constraint problems from other formalisms or domains beyond MiniZinc.

**Computational cost underspecified:** While "reasonable computational budgets" are claimed, the actual cost of the LLM agent workflow isn't clearly reported. How many LLM calls? What's the total wall-clock time? Cost in API calls?

**Missing ablations:** No ablation on key design choices. What if you skip the error correction loop? What if you use simpler prompts? How sensitive is performance to the JSON schema quality?

**Prompting sensitivity unclear:** The appendix shows low variance across 3 runs for problem-specific extractors, but this is limited. What happens with different prompt formulations? How much prompt engineering was needed to get this to work?

**Unclear baseline representativeness:** The paper uses 5 solvers (Gurobi, CPLEX, SCIP, Gecode, OR-Tools) but doesn't justify whether these represent the state-of-the-art or are the most relevant for the problem types tested. More context on solver selection would help assess the practical significance of the results.

**Writing quality and presentation issues:** Multiple presentation problems hurt clarity: (1) Abstract doesn't adequately specify what kinds of tasks the method is applied to, or what specific datasets (MiniZinc) are considered in the paper. (2) Intro makes broad claims without citations (e.g., lines 40-42 about companies unable to use algorithm selection). (3) Intro lacks concrete examples of the CSPs being addressed. (4) Section 3 reads like technical documentation rather than research methodology. (5) Writing quality notably degrades in Section 4 with unclear exposition.

**Reproducibility concerns:** While a reproducibility statement is included, the reliance on closed-source LLMs (o4-mini, Claude Sonnet 4) and proprietary solvers limits true reproducibility. What happens when these model versions are deprecated? Including at least one open-source LLM (e.g., Llama, Qwen, Mistral) in the evaluation would strengthen the empirical argument and improve long-term reproducibility.

**Questions:**

1. Can you clarify your novelty claim? There's prior work on LLMs for automated feature engineering (CAAFE [2], AS-LLM [3], surveys [6]). What specifically is novel beyond applying existing LLM-for-feature-engineering techniques to the CP/algorithm selection domain?

2. Can you provide concrete examples in the intro of specific CSPs you're addressing (e.g., vehicle routing, scheduling) to help readers understand the problem space?

3. How does your approach compare to recent neural approaches [4, 5] in terms of both performance and computational cost? Can you provide head-to-head comparisons on the same benchmark? What's the total computational cost (wall-clock time, LLM API calls, dollars) for generating extractors for all 227 problems?

4. How representative are your 5 baseline solvers of the current state-of-the-art for MiniZinc problems? Why were these specific solvers chosen?


## References

[1] Roberto Amadini, Maurizio Gabbrielli, and Jacopo Mauro. "An enhanced features extractor for a portfolio of constraint solvers." In Symposium on Applied Computing, SAC 2014, pp. 1357–1359. ACM, 2014.

[2] Noah Hollmann, Samuel Müller, and Frank Hutter. "Context-Aware Automated Feature Engineering." In International Conference on Machine Learning (ICML), 2024.

[3] Large Language Model-Enhanced Algorithm Selection: Towards Comprehensive Algorithm Representation. arXiv preprint, January 2024. https://arxiv.org/html/2311.13184v2

[4] Alessio Pellegrino, Özgür Akgün, Nguyen Dang, Zeynep Kiziltan, and Ian Miguel. "Transformer-Based Feature Learning for Algorithm Selection in Combinatorial Optimisation." In 31st International Conference on Principles and Practice of Constraint Programming (CP 2025), volume 340 of LIPIcs, pp. 31:1–31:22, 2025.

[5] Zhanguang Zhang, Didier Chételat, Joseph Cotnareanu, Amur Ghose, Wenyi Xiao, Hui-Ling Zhen, Yingxue Zhang, Jianye Hao, Mark Coates, and Mingxuan Yuan. "Grass: Combining graph neural networks with expert knowledge for SAT solver selection." In Proceedings of the 30th ACM SIGKDD Conference on Knowledge Discovery and Data Mining, KDD 2024, pp. 6301–6311. ACM, 2024.

[6] Survey: "Large Language Models for Constructing and Optimizing Machine Learning Workflows." arXiv preprint, November 2024. https://arxiv.org/html/2411.10478v1

---

> ### Author Response · Authors · 2025-11-21
>
> Thank you for the thorough review. The revised manuscript strengthens the empirical evaluation, positioning, and accessibility based on your feedback.
>
> **1. Novelty & Related Work (W4: Overstated novelty claim)**
>
> We address a structural gap in the literature identified by your review: while CAAFE optimizes tabular data and GRASS embeds CNF, no framework existed for synthesizing extractors from high-level symbolic constraints (MiniZinc). Our contribution advances LLM-based feature engineering by targeting symbolic constraint models with executable graph builders. We introduce LLM-synthesized code that builds typed bipartite graphs from MiniZinc models, enabling interpretable algorithm selection across 227 problems spanning 17 years.
>
> Section 2 positions this relative to CAAFE (tabular), AS-LLM (representation learning), GRASS (SAT/CNF), and Pellegrino et al. (transformers). Key distinction: symbolic models → executable code → typed graphs → interpretable features.
>
> **2. Baseline Comparisons**
>
> Head-to-head comparison with trans2feat (Pellegrino 2025) in Appendix C: LLM2feat achieves 84.6% vs 77.8% on FLECC (+6.8pp), and 64.0% vs 59.7% on Car Sequencing (+4.3pp) across all 116 trans2feat variants. LLM-synthesized features achieve competitive accuracy while preserving interpretability, a practical advantage for deployment.
>
> GRASS (Zhang 2024) operates on CNF/SAT with GNN embeddings; our framework targets MiniZinc with richer constructs (global constraints, arrays, objectives). Direct comparison is not feasible due to different input formalisms.
>
> **3. Cost & LLM Dependence**
>
> The synthesis cost ($0.27, 210s per extractor) is negligible compared to solver runtime budgets (20-minute timeouts). Per-instance extraction (<1 min) enables real-time algorithm selection. This efficiency makes the approach practical for production use (Appendix D: 30-run statistics, ~17 iterations average).
>
> Appendix A.10 and Appendix C report experiments with multiple LLMs. Frontier agentic models (o4-mini, Claude Sonnet) reliably execute the protocol (10/10 trials), while current open-source models (Llama 3.3, DeepSeek R1) do not yet support the multi-step verification required (0/10). This establishes a concrete benchmark for evaluating agentic code synthesis capabilities, a key research direction. Among successful models, extractor variations produce <2% accuracy fluctuation.
>
> **4. Ablations**
>
> Section 5.1 and Appendix A.10 report ablations: removing any component (checking, fixing, or verification) → 0% success rate. Each component is essential for reliable code synthesis.
>
> **5. Venue Fit & Accessibility**
>
> While the domain is CP, the core technique (transforming symbolic constraints into graph representations via executable code) generalizes to any formalism with typed variables: SAT, SMT, MIP, PDDL. This is a representation learning contribution suitable for ICLR. Introduction includes VRP example; Section 2 provides formal AS definitions.
>
> **6. Solver Portfolio Representativeness**
>
> Section 5.1 justifies solver selection: Gurobi/CPLEX (commercial MIP), SCIP (hybrid CP-MIP), Gecode (CP baseline), OR-Tools (MiniZinc Challenge gold medalist 2023-2025). The portfolio spans solver paradigms, ensuring heterogeneous behavior for meaningful algorithm selection.
>
> **7. Writing & Presentation**
>
> Revised: Abstract specifies datasets/task, Introduction includes VRP walkthrough, Section 2 provides formal background, and figures are clarified.
>
> **8. Reproducibility**
>
> Generated extractors are deterministic Python scripts; re-running experiments requires no further LLM calls. The protocol establishes a benchmark for multi-step agentic reasoning. The solver portfolio includes open-source tools (SCIP, Gecode, OR-Tools).
>
> **Questions:**
>
> **Q1 (Novelty):** Three specific contributions distinguish our work: (i) the LLM→code→graph architecture for symbolic constraints, (ii) the universal bipartite graph representation, and (iii) the 227-problem evaluation benchmark. See Section 1 and "Novelty & Related Work" above.
>
> **Q2 (Examples):** The revised manuscript includes a VRP walkthrough in the introduction and detailed case studies for VRP/CS/FLECC in Section 5.4.
>
> **Q3 (Baselines & cost):** We outperform trans2feat by +6.8pp (FLECC) and +4.3pp (Car Sequencing), see Section 2 and "Baseline Comparisons" above. Total synthesis cost: ~$61 and ~13h for all 227 extractors; per-instance extraction <1 min.
>
> **Q4 (Solvers):** Our portfolio spans solver paradigms: Gurobi/CPLEX (commercial MIP), SCIP (hybrid CP-MIP), Gecode (CP baseline), and OR-Tools (recent MiniZinc Challenge winner). Justification in Section 5.1.

---

### Official Review · Reviewer_35BG · 2025-10-31

**Soundness:** 3
**Presentation:** 3
**Contribution:** 2
**Rating:** 4
**Confidence:** 3

**Summary:**

This work proposes an LLM-based automated feature extraction framework to address the algorithm selection challenge in combinatorial optimization problems. This method uses LLMs to generate executable Python scripts as feature extractors, combined with symbolic graph structure analysis, implementing a gray-box paradigm to balance automation and interpretability. Experiments on 227 combinatorial problem classes validate the method's effectiveness.

**Strengths:**

1. Combines the program synthesis capability of LLMs with classical feature engineering, avoiding the pitfalls of black-box models.
2. A well-written manuscript with clear content organization.

**Weaknesses:**

1. This manuscript claims that "In general, we can use all LLM models as the backend of the agent. But we find our framework is not sensitive to the potential hallucination of LLM." However, the method is evaluated with a limited set of LLMs, making it difficult to support this claim.
2. Lack of comparative methods: The method proposed in this work is only compared with a method introduced before 2014. It is recommended to include more recent methods for comparison to enhance the persuasiveness of the experiments. Additionally, the paper lacks a thorough survey of related work. It is suggested to include a detailed and qualitative comparison with highly relevant works in the related work section.
3. Lack of detailed cost analysis: For example, the total time spent on feature extraction using LLM2feat is not provided.
4. Limited novelty: This work resembles more of a technical report than an academic paper, lacking sufficient innovation and the insights it should provide to the readers.

**Questions:**

Please refer to the weaknesses.

---

> ### Author Response · Authors · 2025-11-21
>
> Thank you for the constructive feedback and for noting the well-written manuscript with clear content organization. Your comments helped us strengthen the empirical evaluation and better position the work within the broader landscape of LLM-based feature engineering. The revised manuscript addresses all four weaknesses you identified with neural baselines, detailed cost analysis, model sensitivity experiments, and strengthened positioning within the LLM feature engineering landscape.
>
> **Novelty Claims & Positioning (W4: Limited novelty)**
>
> We introduce a method for extracting structural features from symbolic constraint models by having LLMs synthesize executable graph-building code. Unlike tabular feature engineering (CAAFE), this requires transforming logical constraints into graph representations. While CAAFE targets tabular data and GRASS targets CNF formulas, we synthesize executable programs that build typed bipartite graphs from MiniZinc models. The contribution sits between white-box (manual features) and black-box (neural embeddings): LLM-synthesized code produces interpretable graph features at scale.
>
> Section 2 positions this contribution relative to CAAFE (tabular data), AS-LLM approaches, GRASS (SAT/CNF), and Pellegrino et al. (transformers). The key distinction is the symbolic intermediate representation: constraint models → executable code → typed graphs → features.
>
> **Baselines (W2: Lack of comparative methods)**
>
> Head-to-head comparison with trans2feat (Pellegrino 2025) on FLECC and Car Sequencing shows LLM2feat achieves 84.6% vs trans2feat's best of ~78% accuracy, while requiring no per-problem retraining (Appendix C, Tables 13-16). This establishes that explicit program synthesis can match the expressivity of latent neural representations (trans2feat) while offering superior interpretability, a practical advantage for deployment.
>
> Our framework targets MiniZinc constraint models with rich constructs (global constraints, arrays, optimization objectives). GRASS (Zhang 2024, source code publicly unavailable) operates on CNF/SAT formulas; the approach could, in principle, be adapted to CNF as another symbolic input language.
>
> **Cost Analysis (W3: Lack of detailed cost analysis)**
>
> The synthesis cost ($0.27, 210s per extractor) is negligible compared to solver runtime budgets (20-min timeouts). Per-instance extraction (<1 min) enables real-time algorithm selection. Full breakdown in Appendix D shows ~17 iterations average with detailed token/cost statistics.
>
> This efficiency makes the approach practical for production algorithm selection pipelines.
>
> **Model Sensitivity & LLM Dependence (W1: LLM hallucination sensitivity)**
>
> Appendix A.10 and Appendix C report experiments with multiple LLMs. Frontier agentic models (o4-mini, Claude Sonnet) reliably execute the check-fix-verify protocol (10/10 success), while current open-source models (Llama 3.3, DeepSeek R1) do not yet support the multi-step verification required (0/10). This establishes a concrete benchmark for evaluating agentic code synthesis capabilities, a key research direction as open models continue to improve.
>
> Among successful models, extractor variations produce <2% accuracy fluctuation, demonstrating that the protocol is model-agnostic once the capability threshold is met.

---

### Official Review · Reviewer_hoLH · 2025-10-31

**Soundness:** 3
**Presentation:** 3
**Contribution:** 2
**Rating:** 4
**Confidence:** 3

**Summary:**

This paper presents a gray-box approach for algorithm selection. Given a feature selection templates, the LLMs are provided with problem data, json schema, and feature selection template (py script), and it is asked to generate a feature extractor to derive 50 standardized features. The evaluation shows that LLM extracted features are diverse, efficient and with high accuracy.

In general, this paper is a nice application and study of how LLM agents can be used in algorithm selection pipeline to extract features. The detailed analysis shows its performance benefit comparing to traditional approaches.

**Strengths:**

- clear formulation of the problem and quite thorough analysis showing benefits
- the graybox approach is a nice intermediate for human AI collaboration.
- simple but effective pipeline

**Weaknesses:**

- As I'm not an expert in algorithm selection problem, it is a bit hard to interpret the significance of the result comparing to work in the field. But from the LLM agent design perspective, the design is quite straightforward. While I can see the application value, I would like to read more about how different designs of LLM agents matters, and some analysis behind why these features are more effective than previous approaches beyond observations of they are better (which is very well presented!)

- The problem formulation (section 2) seems a bit disconnected overall, as an non-expert in AS domain, the formulation shows the overall task, but not much details on how features affect AS results. Some more details in this aspect may better highlight the success of LLM-based gray-boxed approach.

**Questions:**

Check the weakness section above - can you provide some qualitative analysis on why LLM extracted features are more effective and what information they leverage made the difference.

---

> ### Author Response · Authors · 2025-11-21
>
> Thank you for the positive assessment and for your interest in the practical impact of LLM-synthesized features. Your questions helped us add concrete examples that illustrate how the framework discovers solver-relevant patterns.
>
> **Significance of Results & Design Clarity (W1: Qualitative analysis on why features are effective)**
>
> The revised manuscript adds:
> - **Section 2**: Intuition linking features to solver performance (how Φ captures constraint graph density, domain sizes, and tightness patterns that correlate with solver behavior)
> - **Section 5.4**: "Qualitative Feature Examples" with VRP case study
> - **Appendix**: Expanded analysis categorizing features by domain
>
> **VRP Case Study - Solver-Aware Features:**
>
> The LLM-synthesized extractor for VRP discovers domain-specific structural properties:
>
> - **Demand distribution**: `avg_demand`, `std_demand`, `skew_demand`, `coeff_var_demand` capture load variability. High variance/skewness favor MIP solvers (better linear relaxations); uniform demand favors CP solvers (spatial propagation).
> - **Depot centrality**: `depot_centrality` (closeness), `depot_bc` (betweenness) measure depot accessibility. Central depots favor CP branching heuristics; eccentric depots favor MIP formulations.
> - **MST metrics**: `mst_total`, `mst_avg_edge`, `mst_std_edge` approximate route-length lower bounds. Instances with tight MST bounds favor MIP (tight LP relaxations); loose bounds favor CP global constraints.
>
> These features capture the structural properties that solvers exploit, explaining the +2.7 percentage-point accuracy gain on VRP. The approach re-discovers and codifies expert intuition: the LLM identifies that 'depot centrality' dictates the effectiveness of branching heuristics, a relationship previously known only to domain experts.
>
> **Check-Fix-Verify Loop Necessity:**
>
> Ablation results (Section 5.1, Appendix A.10): removing any component (checking, fixing, or verification) → 0% success rate. Each step is essential for reliable code synthesis.
>
> **10% Accuracy Gain Context:**
>
> The 10pp gain (48.6% → 58.8%) confirms a hypothesis: that structural signals lost in translation to flat feature vectors can be recovered via program synthesis. LLM-synthesized features extract this information more effectively than hand-engineered syntax features (mzn2feat), while maintaining interpretability advantages over neural embeddings (trans2feat). The gains are consistent across the 227-problem benchmark suite.
>
> The field has seen minimal progress since 2014, as the most recent algorithm selection competition took place in 2017 [1].
>
> [1] Algorithm selection competition: https://www.coseal.net/open-algorithm-selection-challenge-2017-oasc/

---

### Official Review · Reviewer_phvD · 2025-11-05

**Soundness:** 3
**Presentation:** 3
**Contribution:** 2
**Rating:** 4
**Confidence:** 3

**Summary:**

The paper leverages LLMs to generate Python scripts as a “gray box” to extract features for combinatorial optimization problems, with the purpose of increasing the interpretability of the feature extractor and reducing human effort. Experimental results on the algorithm selection problem demonstrate the effectiveness and improved feature quality of the proposed method.

**Strengths:**

1. The proposed framework is reasonable for improving interpretability.
2. Experimental results show the improved quality of the extracted features and increased accuracy on the optimization problem to which the features contribute.

**Weaknesses:**

1. The performance of the feature extractor is highly dependent on the LLM, which is expected to contain sufficient domain knowledge of the problem and the reasoning ability to select significant features based on the verification results. Without post-training, the range of solvable problems could be limited.
2. Some details of the pipeline are unclear. From my understanding, the LLM agent is used to generate a script extractor, which is then used to extract features from the problem data to train a solver predictor. During LLM generation, does it receive feedback from the verification results or generate a certain number of candidates for further selection? It seems that the former strategy is used in Figure 1, while the latter is used in Figure 2. If the LLM receives feedback to refine the script, what kind of feedback is it, and how is it provided in the prompt?

**Questions:**

1. One purpose of generating the script extractor from the LLM is to increase interpretability. Adding some script examples in the paper or appendix could enhance the illustration of this contribution.
2. Are there any trade-offs between accuracy and interpretability? For example, if the LLM is directly used to extract features, would the feature quality be better than generating the script extractor?

---

> ### Author Response · Authors · 2025-11-21
>
> Thank you for the constructive feedback and for noting that the framework is reasonable for improving interpretability. Your questions helped us clarify the iterative refinement protocol and better explain the dual-mode operation of the framework.
>
> **LLM Dependence & Domain Knowledge Requirements (W1: Performance highly dependent on LLM)**
>
> The framework demonstrates two modes of operation: domain-aware synthesis (discovering solver-relevant patterns like VRP demand concentration) and domain-agnostic synthesis (pure syntactic graph construction from MiniZinc keywords: `var`, `constraint`, `array`). Both succeed without fine-tuning, demonstrating the generality of the check-fix-verify protocol. Frontier agentic models (o4-mini, Claude Sonnet) reliably support the protocol; this establishes a capability benchmark for evaluating code synthesis in future LLMs (Appendix A.10).
>
> **Feedback Loop Clarification (Figure 1 vs Figure 2):**
>
> Section 4.2 clarifies that both frameworks (Figures 1 and 2) employ the same check-fix-verify loop. The iterative refinement protocol:
>
> 1. LLM proposes a Python script
> 2. Execute the script on a validation instance
> 3. If error → pass raw `stderr` trace to LLM (e.g., `ZeroDivisionError at line 47`)
> 4. LLM refines the same script based on the error location
> 5. Repeat until verification passes
>
> We utilize deterministic execution feedback rather than LLM self-reflection. The agent receives the precise Python traceback (e.g., `ZeroDivisionError`), grounding the correction in ground-truth runtime behavior.
>
> This single-script iterative approach (~17 iterations average) enables reliable code synthesis from frontier LLMs. Detailed prompts are in Appendix A.1-A.2 and supplementary materials.
>
> **Feedback Example:**
>
> ```python
> # Iteration 1: LLM generates script
> graph_density = num_edges / num_nodes  # Bug: potential division by zero
>
> # Verification → "RuntimeError: division by zero (line 47)"
>
> # Iteration 2: LLM receives error, generates fix
> graph_density = num_edges / num_nodes if num_nodes > 0 else 0.0  # Passes
> ```
>
> **Script Example (FLECC Feature Extractor):**
>
> ```python
> def main():
>     instance_data = input_data()
>     c = instance_data.get('nbCharacter')     # Alphabet size
>     L = instance_data.get('codeWordLength')  # Word length
>     dist_matrix = instance_data.get('dist')  # Distance matrix
>
>     # Statistical features
>     mat_mean = dist_matrix.mean()
>     mat_std = dist_matrix.std()
>
>     # Graph-based features
>     G = build_constraint_graph(c, L)
>     graph_density = nx.density(G)
>     graph_clustering = nx.average_clustering(G)
>
>     results = {"alphabet_size": c, "constraint_density": graph_density, ...}
> ```
>
> Features are semantically transparent and debuggable (Appendix A.2 and supplementary materials contain complete extractors).
>
> **Interpretability vs Accuracy Trade-off:**
>
> This design decouples reasoning (LLM) from computation (Python), eliminating arithmetic hallucinations common in direct-prompting approaches. The executable script approach combines LLM reasoning with deterministic computation (NumPy/NetworkX), ensuring numerically exact features at scale. The approach maintains interpretability (human-readable code with explicit feature names like `depot_centrality`, `mst_total`) while guaranteeing numerical reliability.

---

### Author Response · Authors · 2025-11-21
**Overall Response**

**Overall Response to Reviewers**

We thank all reviewers for their engagement with our work. The discussion enabled us to strengthen the empirical evaluation and better communicate the framework's contributions.

This work introduces LLM-synthesized, interpretable graph extractors for algorithm selection on symbolic constraint models. The approach achieves 58.8% accuracy vs 48.6% baseline on 227 MiniZinc problems, and outperforms neural methods by +6.8pp (FLECC) and +4.3pp (Car Sequencing) while maintaining full interpretability.

The revised manuscript now includes:

• **Related Work (Section 2):** Comprehensive coverage of CAAFE (tabular data), AS-LLM approaches (representation learning), GRASS (SAT/CNF), and transformer-based feature learning (Pellegrino et al.). Positions our contribution: symbolic constraint models → executable code → typed graphs → interpretable features.

• **Neural Baseline Comparison:** Head-to-head comparison with trans2feat on overlapping benchmarks. LLM2feat outperforms all 116 trans2feat variants: 84.6% vs 77.8% on FLECC (+6.8 pp), 64.0% vs 59.7% on Car Sequencing (+4.3 pp). This demonstrates competitive accuracy while preserving interpretability, a practical advantage for deployment. Full results in Appendix C.

• **Cost & Model Sensitivity:** Generation cost: $0.27, 210s per extractor (~17 iterations). Per-instance extraction: <1 min vs 20-min solver timeout. Frontier agentic models (o4-mini, Claude Sonnet) reliably execute the protocol (10/10); current open-source models (Llama 3.3, DeepSeek R1) do not yet support the multi-step verification (0/10). This establishes a benchmark for evaluating agentic code synthesis capabilities. Full statistics in Appendices C, D and A.10.

• **Qualitative Feature Analysis:** Section 5.4 adds VRP case study with real feature names (`avg_demand`, `depot_centrality`, `mst_total`) showing how LLM-discovered features capture solver-relevant structural properties.

• **Pipeline Clarity:** Aligned Figures 1–2 (added feedback arrow to Figure 2), updated Section 4.2 to clarify the iterative refinement protocol.

• **Reproducibility:** All generated extractors and prompts are included in the supplementary materials. Generated extractors are deterministic Python scripts; re-running experiments requires no further LLM calls.

This work shows that LLMs can synthesize code that transforms symbolic models into interpretable ML features. The approach generalizes to other formalisms (SAT/SMT/MIP/PDDL) and opens research directions in combining generative AI with symbolic reasoning.

---

### Meta-Review · Area_Chair_GQDM · 2026-01-12

**Summary:**

The reviewers raised many important concerns and unfortunately the authors and reviewers were unable to engage in a discussion before the unexpected discussion freeze. Many of the concerns raised by the reviewers seem to be related to a struggle in understanding the significance of the problem and limited information for evaluating the novelty of the proposed solution. The authors have tried to address this by providing additional context and a deeper discussion of prior work in the paper. However, it is unlikely that these updates would lead to significant changes in the reviewers’ scores. In particular, a key assertion from the authors in response to the reviewers concerns about the significance of the contribution is that the ‘approach generalizes to other formalisms (SAT/SMT/MIP/PDDL)’. However, this statement is provided without any experimental justification. Additional experiments on at least one of these formalisms would have strengthened the paper. The additional prior work suggested by reviewers has also moderated some of the novelty claims in the original manuscript, and this necessitates the need for additional experiments or explorations of other formalisms to support this paper.

**Reviewer Concerns:**

The primary concerns seems to be around novelty and clarity in the paper's positioning. The authors have addressed positioning with changes to the manuscript, and additional discussion of related work. Criticisms about novelty are relatively unanswered, especially since the primary claim of generalizing to other formalisms is provided without evidence.

**Reviewer Scores:**

None of the reviewers are likely to have changed their scores.

---

### Decision · Program_Chairs · 2026-01-26

Reject